# Asymmetric pendrin homodimer reveals its molecular mechanism as anion exchanger

Qianying Liu [1,8], Xiang Zhang[1,8], Hui Huang[1,8], Yuxin Chen [2,3,4,8], Fang Wang[2,3,4], Aihua Hao[1], Wuqiang Zhan[1], Qiyu Mao[1], Yuxia Hu[1], Lin Han[1], Yifang Sun[1], Meng Zhang[1], Zhimin Liu[1], Geng-Lin Li[2,3,4], Weijia Zhang [1], Yilai Shu [2,3,4,5] ✉, Lei Sun [1,6,7] ✉ & Zhenguo Chen [1,6,7] ✉

Pendrin (SLC26A4) is an anion exchanger expressed in the apical membranes of selected epithelia. Pendrin ablation causes Pendred syndrome, a genetic disorder associated with sensorineural hearing loss, hypothyroid goiter, and reduced blood pressure. However its molecular structure has remained unknown, limiting our understanding of the structural basis of transport. Here, we determine the cryo-electron microscopy structures of mouse pendrin with symmetric and asymmetric homodimer conformations. The asymmetric homodimer consists of one inward-facing protomer and the other outward-facing protomer, representing coincident uptake and secretion- a unique state of pendrin as an electroneutral exchanger. The multiple conformations presented here provide an inverted alternate-access mechanism for anion exchange. The structural and functional data presented here disclose the properties of an anion exchange cleft and help understand the importance of disease-associated variants, which will shed light on the pendrin exchange mechanism.

Over 5% of the world population has hearing loss problems (according to WHO data) and among these 3–5% is attributed to *SLC26A4* disease-associated mutations varying by ethnicity[1–3]. Encoded by the gene *SLC26A4*, pendrin belongs to the solute carrier 26 (SLC26) family[4]. Originally, pendrin was identified as a sodium-independent transporter of Cl⁻ and I⁻ in heterologous cell expression systems (*Xenopus laevis* oocytes or insect Sf9 cells)[5]. Later, it was widely considered as a sodium-independent electroneutral $Cl^-/HCO_3^-$ and $Cl^-/I^-$ exchanger expressed in the apical membrane of the inner ear, thyroid, and kidney epithelial cells[6–9] (Fig. 1a). In humans, pendrin ablation causes the genetic disorder Pendred syndrome[10], associated with sensorineural hearing loss, hypothyroid goiter, and reduced blood pressure[9,11].

Numerous genetics studies on Pendred syndrome patients and various mouse models have provided insight about pendrin. In the inner ear, pendrin was observed in the regions where endolymphatic fluid reabsorption occurs[6]. Briefly, pendrin is expressed in the epithelial cells of cochlear spiral prominence, endolymphatic sac, saccule, utricle, and ampulla[12]. The transport and exchange function of pendrin is taken to be critical in regulating the composition and pH stability of the inner ear's endolymph. Pendrin knockout in mice leads to the development of an enlarged vestibular aqueduct (EVA), causing hearing impairment[13], and affects the acid-base balance of the inner ear which causes sensorineural hearing loss[14,15]. In the kidney, pendrin localizes to type B and non-A, non-B intercalated cells (ICs) within the connecting tubule (CNT) and cortical collecting duct (CCD), mediating renal Cl⁻ absorption and $HCO_3^-$ secretion[7,9]. Within these apical plasma membranes, the abundance and activity of pendrin are significantly stimulated by angiotensin II and aldosterone, therefore it would be beneficial to correct the

[1]The Fifth People's Hospital of Shanghai and Institutes of Biomedical Sciences, Fudan University, Shanghai 200032, China. [2]ENT Institute and Department of Otorhinolaryngology, Eye & ENT Hospital and State Key Laboratory of Medical Neurobiology, Fudan University, Shanghai 200031, China. [3]MOE Frontiers Center for Brain Science, Fudan University, Shanghai 200031, China. [4]NHC Key Laboratory of Hearing Medicine, Fudan University, Shanghai 200031, China. [5]Institutes of Biomedical Sciences, Fudan University, Shanghai 200032, China. [6]Shanghai Institute of Infectious Disease and Biosecurity, Shanghai 200032, China. [7]Shanghai Key Laboratory of Medical Epigenetics, Shanghai 200032, China. [8]These authors contributed equally: Qianying Liu, Xiang Zhang, Hui Huang, Yuxin Chen. ✉e-mail: yilai_shu@fudan.edu.cn; llsun@fudan.edu.cn; zhenguochen@fudan.edu.cn

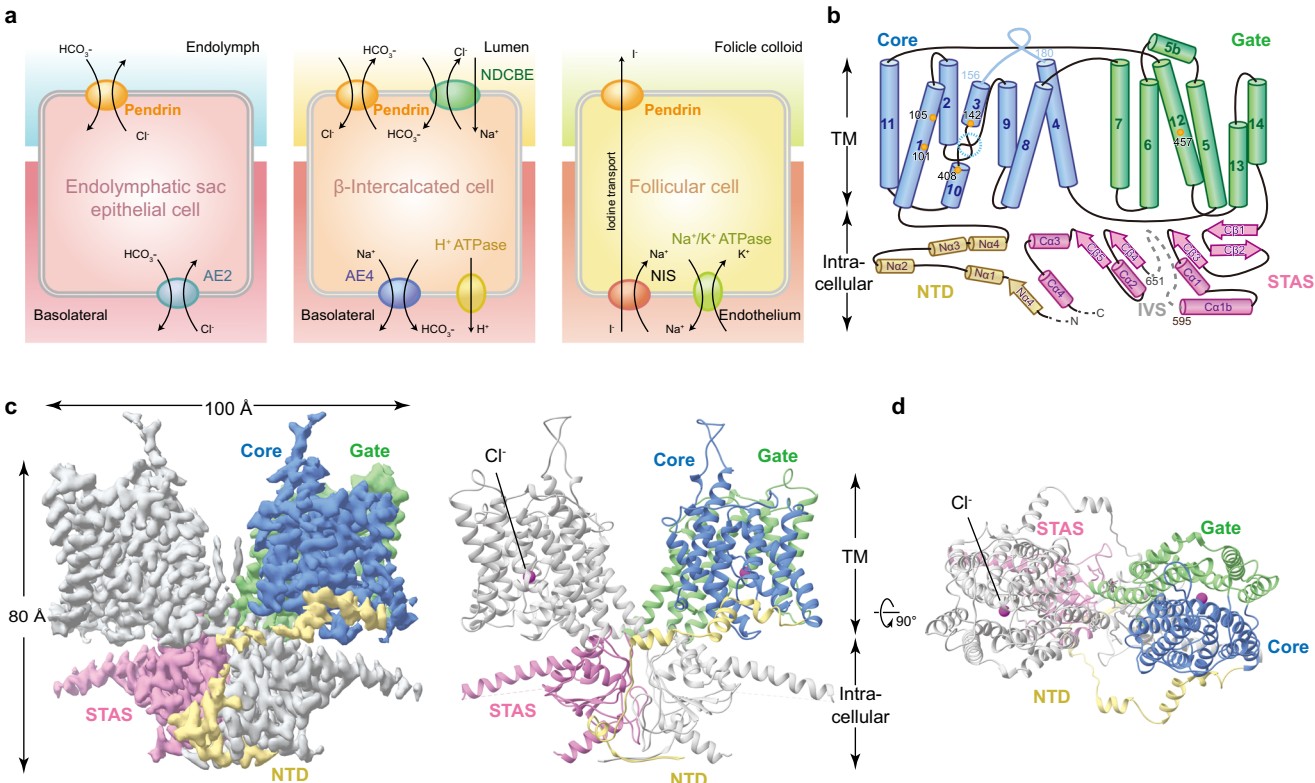

**Fig. 1 | Cellular functions and cryo-EM structure of Pendrin. a** Cellular functions of pendrin. In the inner ear, pendrin mediates the exchange of $HCO_3^-$ and $Cl^-$; in β-intercalated cells, it participates in urinary $HCO_3^-$ excretion and tubular $Cl^-$ reabsorption; and in the thyroid, pendrin is involved in apical $I^-$ transport. NIS, sodium iodide symporter. **b** Topology of Pendrin. The N-terminal (NTD), core region, gate

region, and STAS domains are colored in yellow, blue, green, and pink, respectively. The anion-binding pocket is marked by a dashed circle in cyan. The key residues around the pocket are labeled. **c** Cryo-EM map and structural model of mouse pendrin-Cl. One protomer is colored in grey, and the other is colored as (**b**). **d** View of c from outside the membrane. The $Cl^-$ are shown as spheres in magenta.

alkalosis and maintain NaCl homeostasis to regulate blood pressure in the kidney[7,16–18]. In the thyroid, pendrin is involved in $I^-$ transport, however, its precise function is still under debate[19].

Despite the importance of pendrin, the exchange mechanism has remained unknown. Here, we investigate the structural and functional properties of mouse pendrin using single particle cryo-electron microscopy (Cryo-EM) and anion exchange assays. We observe that the structures of pendrin vary in the presence of different anions, including three distinct states: a symmetric inward-open dimer, a symmetric outward-open dimer, and a characteristic asymmetric dimer composed of one inward-open protomer and one outward-open protomer. Moreover, clear anion densities are observed in both inward-open and outward-open protomers. This unique asymmetric homodimer architecture aids understanding of the pendrin exchange process as an inverted alternate-access mechanism, with one protomer transporting in- and another transporting out- to provide the structural basis to describe electroneutral anion exchange.

## Results and discussion

### Pendrin forms a symmetric homodimer in the presence of $Cl^-$

The full-length mouse pendrin with an N-terminal affinity tag was expressed in HEK293E cells and purified using glycol-diosgenin (GDN) in the presence of 150 mM NaCl. The purified protein was concentrated to 1.7 mg/mL and immediately used for cryo-EM grid freezing and subsequent data collection. Cryo-EM data processing showed that pendrin forms a dimer as other family members[20–25] (Supplementary Fig. 1). The overall dimer structure was determined to 3.4 Å without symmetry, showing two protomers are almost identical. C2 symmetry was then applied for further refinement, which improved the reconstruction map to 3.3 Å resolution, allowing the unambiguous

interpretation of the cryo-EM density map to the atomic model (Supplementary Fig. 2a).

Pendrin purified in 150 mM NaCl (Pendrin-Cl) forms a domain-swapped homodimer (Fig. 1c, d). Each protomer is composed of an N-terminal domain (NTD) (residues 1-79), a transmembrane domain (TMD) (residues 80-515), a sulfate transporter and anti-sigma factor antagonist domain (STAS) (residues 516-735) containing the intervening sequence (IVS) (residues 596-650) and a C-terminal domain (CTD) (residues 736-780) (Fig. 1b). Residues 1-17, 596-650 and 738-780 are missing in the model due to the flexibility. The NTD and STAS domains from each protomers interchange and form a dimeric knob at the cytoplasmic side of the plasma membrane. On the other side, a spiky loop from each protomer sticks out from the membrane. The TMD follows the so-called UraA fold[26,27], with 14 TMDs divided into two inverted repeats representing the core (helices TM 1-4 and 8-11) and gate regions (helices TM 5-7 and 12-14) (Figs. 1b and 2a).

### The anion-binding site between the core and gate regions within the TMD

In pendrin-Cl structure, an unambiguous $Cl^-$ density is observed and defines the anion-binding site (Fig. 2d, e). The $Cl^-$ is embraced in the cavity between two short transmembrane helices TM3 and TM10, which is positioned at the apex of the inward-open intracellular vestibule (Fig. 2b, c).

The $Cl^-$ is mainly coordinated by the TM3-TM10 dipoles (Fig. 2c). S408 and Y105 directly interact with $Cl^-$ by forming ionic bonds (Fig. 2d). The side chains of Y105 and Q101 form the partially positive charged binding pocket, which is quite common across the family[21–23,28]. In addition, the positively charged residue R409 indirectly stabilizes $Cl^-$ by interacting with the nearby Q101, which in turn

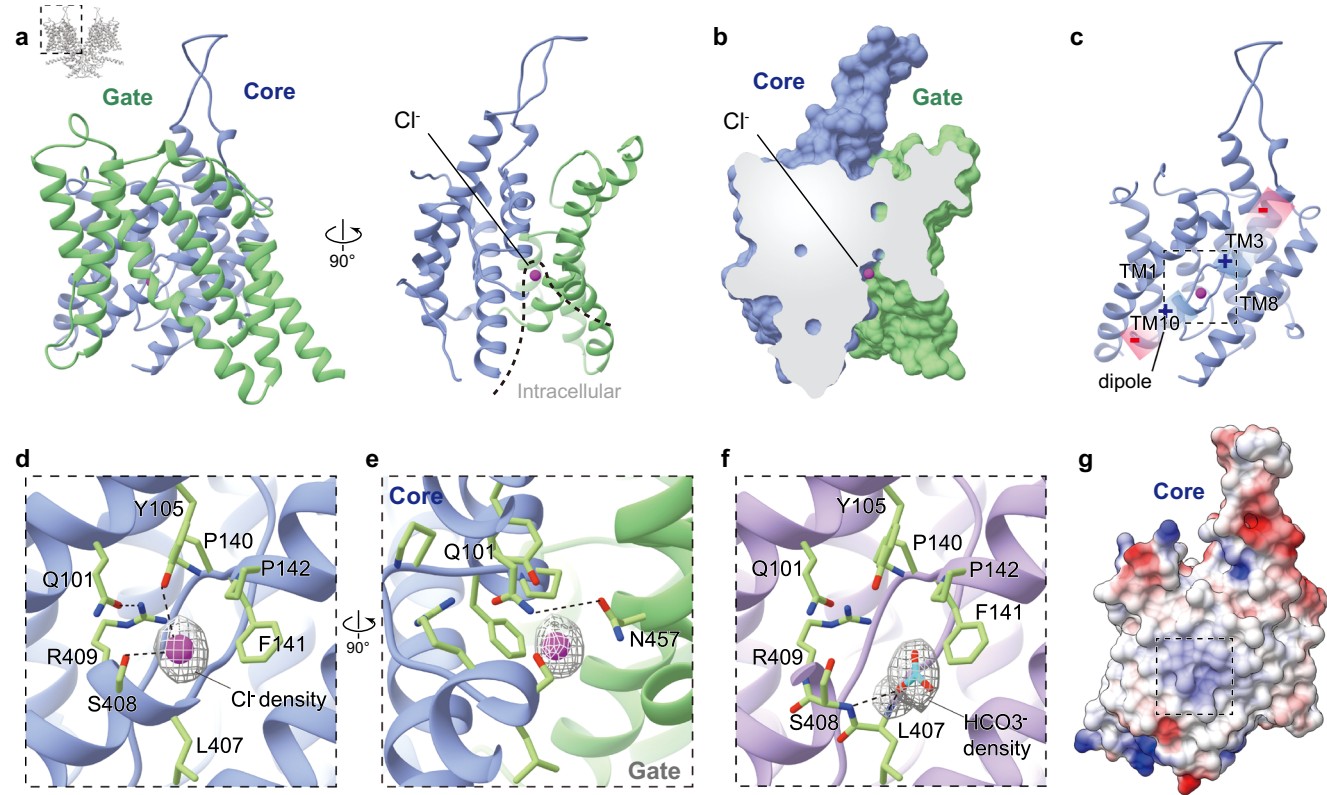

**Fig. 2 | The TMD and anion binding pocket of pendrin. a** Structural model of pendrin-Cl transmembrane domain (TMD) shown in ribbon representation. Core and gate regions are colored in lime green and dodger blue, respectively. The purple sphere indicates bounded Cl⁻, and the dotted line describes the intracellular vestibule. **b** Section of TMD surface model. **c** Structural model of the core region of pendrin-Cl. Gradient rectangles with charge symbols indicate the helical dipoles of TM3 and TM10. **d** Details of the anion-binding pocket of pendrin-Cl. The density representing Cl⁻ is shown in the light grey mesh. Interactions of Y105 and S408 side chains proximal to the pocket are shown by the dashed line. The hydrogen bond between Q101 and R409 is shown by the dashed line. **e** A 90° rotated view of d showing the hydrogen bond between Q101 and N457. **f** Details of the anion-binding pocket of pendrin-HCO₃. The density representing HCO₃⁻ is shown in the light grey mesh. Interactions of L407 and S408 backbone proximal to HCO₃⁻ are shown by the dashed line. **g** Electrostatic potential surface of the core region of pendrin-Cl.

interacts with S408. A406, L407, and three sequential hydrophobic residues P140, F141, and P142 hallmark the partial hydrophobic feature of the pocket (Fig. 2g). F141 and P142 form pi-pi stacking interactions. Additionally, N457 from the gate region forms a hydrogen bond with Q101 from the core region, which also helps to stabilize the conformation of the core and gate regions (Fig. 2e).

Around the anion-binding site, several disease-associated missense variants have been identified in patients and recorded in Deafness Variation Database [https://deafnessvariationdatabase.org][29]. Firstly, the Cl⁻ interacting residues Y105 and S408 have four missense variants (Y105C, S408Y, S408P, and S408F)[29], among which S408F mutation almost abrogated transport activity[30,31]. In addition, R409 inside of the pocket has four disease-associated mutations (R409L, R409P, R409C, and R409H)[29], among which R409H showed a reduction of exchange and transport function[30,32,33]. Though histidine does not abolish the positive charge, it alters the multiplex interaction and charge environment mediated by R409. Hydrophobic A406, P140, F141 and P142 residues also have a few missense variants (A406T, P140H, F141L, F141S, P142R and P142L)[29]. P140H was reported to cause a loss of Cl⁻/I⁻ exchange activity[34] and P142R caused a loss of Cl⁻/HCO₃⁻ exchange activity[35]. P142L induced Cl⁻/I⁻ exchange reduction[36]. In addition, N457 located at the gate region also has four mutations (N457D, N457Y, N457I, and N457K)[29], and N457K reduced both Cl⁻/I⁻ and Cl⁻/HCO₃⁻ exchange[30,37]. Taken together, the local pocket architecture is crucial for anion binding and therefore important for pendrin's anion transport and exchange function.

## HCO₃⁻ binds the same pocket as Cl⁻

To obtain HCO₃⁻ bound Pendrin (Pendrin-HCO₃), pendrin in Cl⁻ buffer was exchanged to a buffer containing HCO₃⁻ and frozen for cryo-EM data collection. The cryo-EM structure was determined to 3.5 Å resolution with C2 symmetry. Similar to pendrin-Cl, pendrin-HCO₃ also forms a symmetric homodimer with inward-open TMD clefts. Anion HCO₃⁻ was clearly observed as well in the binding pocket, which is about 2 Å away from Cl⁻ binding position (Fig. 2f). Though Y105 is too far to interact with, HCO₃⁻ forms extra hydrogen bonds with the backbone nitrogen of S408 and L407.

## Pendrin forms an asymmetric homodimer in the presence of two different anions

In the presence of a single anion, Cl⁻ or HCO₃⁻, only inward-open TMD cleft was observed in our cryo-EM structures. The inward-open state may represent the most thermodynamically stable conformation but is not favored for anion transport and exchange. In physiological circumstances, pendrin is exposed to a mixture of different anions and could bind either anion (of the exchange pair) at both inward-open and outward-open states. Therefore, we investigated pendrin in a buffer containing two different anions, such as Cl⁻/HCO₃⁻, Cl⁻/I⁻, or HCO₃⁻/I⁻ anion pairs.

Pendrin purified in 100 mM NaCl was mixed with the same volume of 300 mM NaHCO₃, resulting in a buffer containing 50 mM NaCl and 150 mM NaHCO₃. The sample was then concentrated and used for cryo-EM data collection. The data was processed without symmetry. Surprisingly, the cryo-EM structural determination of pendrin in the

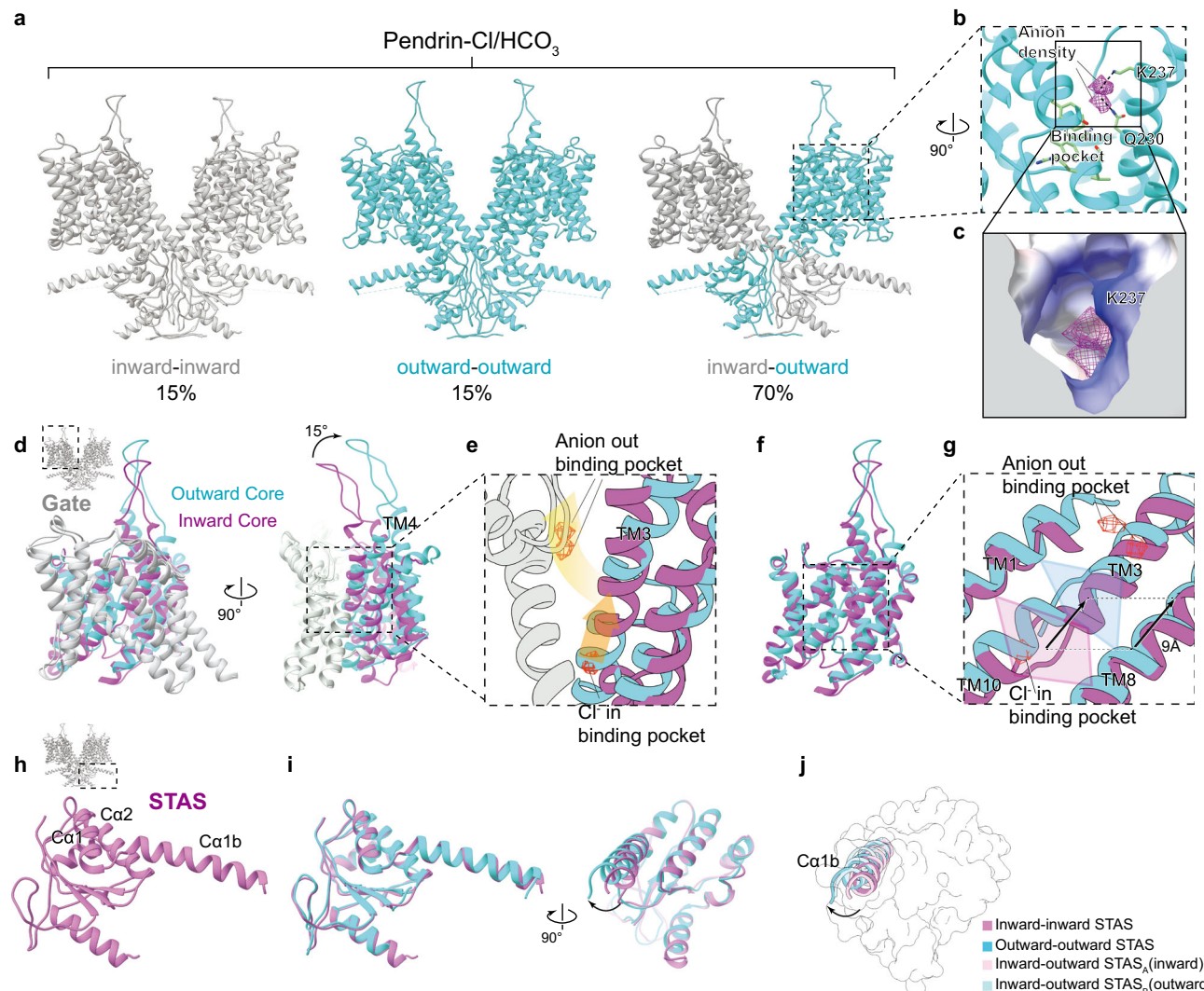

**Fig. 3 | Comparison of inward-open and outward-open states of pendrin.**
**a** Three conformations of pendrin-Cl/HCO$_3$. **b** Details of the open cavity of TMD in outward-open protomer. Densities representing two anions are shown in mesh mode. Interactions of Q230 and K237 proximal to the densities are shown. **c** Section of the open cavity of TMD in outward-open protomer, shown as the electrostatic potential surface. **d** Structural superimposition of the TMDs of inward (orchid and grey) state and outward state (cyan and grey) of pendrin. **e** Details of d. Densities representing anions are shown in mesh mode. Gradient arrows indicate the pathway of anion from the inward-state binding pocket to the outward-state binding pocket, and finally to the open cavity. **f** The core region of (**b**). **g** Details of f. As the core region rotates, the anion-binding pocket indicated by the translucent tetragon translates 9 Å towards the extracellular side. **h** STAS domain of inward state of pendrin-Cl/HCO$_3$. **i** Structural superimposition of the STAS domains of inward (orchid) and outward state (cyan). **j** Differences of the helix Cα1b among four protomers. Inward-inward (orchid), outward-outward (cyan), inward-outward protomer A(pink), and inward-outward protomer B (light blue).

presence of Cl$^-$/HCO$_3^-$ (pendrin-Cl/HCO$_3$) mixture revealed three distinct states (Fig. 3a). Only ~15% percent particles remained symmetric inward-open state. Another 15% percent of particles represent the symmetric outward-open state in which the cavity of the binding pocket is connected to the outside. This outward-open conformation of TMD is similar to that of SLC4A1 (anion exchanger 1, AE1) TMD crystal structure which is stabilized in an outward state by antibody[27]. Interestingly, most particles (~70%) form asymmetric homodimers composed of one inward-open protomer and the other outward-open protomer. In this asymmetric dimer, a clear anion density was observed in the inward-open protomer, which is assigned to the identical Cl$^-$ occupying position as that of pendrin-Cl symmetric structure (Supplementary Fig. 3c). Notably, in the outward-open protomer, there were two individual densities identified in the cavity above the binding pocket (Fig. 3b). The open cavity forms a positively charged pocket that helps to recruit anions from the outside (Fig. 3c) and the side chains of K237 and Q230 could interact with the upper and lower anion density, respectively (Fig. 3b). These two densities are

about 4.2 Å from each other, that is too close to accommodate two anions at the same time (Supplementary Fig. 2b). Therefore, these two densities most likely reveal two binding modes of anion to the pocket, representing two moments of the anion transport process.

Similarly, the pendrin sample in the anion pair of Cl$^-$/I$^-$ (1:3) (pendrin-Cl/I) was prepared and used for structure determination. Symmetric inward-open and asymmetric homodimers were observed with approximate percentage of 75 and 25%, respectively. Finally, the structures of pendrin in HCO$_3^-$/I$^-$ (1:3) (pendrin-HCO$_3$/I) was determined, which also revealed two distinct states: 75% symmetric inward-open and 25% asymmetric states (Supplementary Fig. 3). After superimposing all structures we resolved, we found very little difference among inward-open structures or among outward-open structures, respectively, with root mean squared deviation (RMSD) no more than 0.5 Å (Supplementary Fig. 7a, b and Supplementary Table 1). Since the combination of Cl$^-$ and HCO$_3^-$ provides the most varieties of conformations, pendrin-Cl/HCO$_3$ structures are used for illustration.

The major conformational difference between the inward-open and outward-open pendrin protomers is the relative position of the gate and the core regions (Fig. 3d). When the gate regions are superimposed, the core region in the outward-open state (versus the inward-open state) rotates roughly by 15° and translates 9 Å towards the extracellular side (Fig. 3e–g). This movement is characteristic in SLC26 family members and pendrin has the largest conformational change among published data[21,23,28]. This conformation change moves the upper binding pocket and carries the anion outside like an elevator and then releases it outwards (Fig. 3e). The outward-open state increases (~15%) the cross-section area of the protomer on the outer leaflet when compared with the inward-open state (Supplementary Fig. 4b). Therefore, in the asymmetric homodimer, this area increase would be compensated by the inverted conformation change of the other protomer.

### The STAS domain modulates pendrin transport function

The STAS domain takes 3 β-strands as the core which is surrounded by 4 α-helices, extending the lateral helix Cα1b to link IVS (Fig. 3h). The starting residues 516–545 of the STAS domain form a long loop region, and it is well resolved due to the interactions with Cβ3, Cα4, and NTD (Fig. 4c). Similar to prestin and SLC26A9, pendrin's dimerization interface is mainly formed by STAS domains. The STAS domains of two protomers contact closely face to face on a relatively flat surface and two NTD's Nβ1s parallel inversely below the STAS domains (Fig. 4a). D724 from STAS's Cα4 forms a hydrogen bond with R24 from NTD's Nβ1 of the same protomer, and this interaction is conserved among SLC family members (Fig. 4c). Several pathogenetic missense variants were identified at D724 and R24[38–41]. In the dimerization interface, S552 forms a hydrogen bond with S666 from the other protomer, which would be destroyed by disease-associated mutation S552I[42] (Fig. 4f). Although the residues are not conserved, high structural similarity allows SLC26 members to form interactions between two protomers. In addition, the STAS of one protomer not only interacts with the STAS of another protomer, but also contacts the TMD of another protomer, as the platform formed by Cα1, Cα1b, and Cα2 stays below the TMD

(Fig. 4a, e). Moreover, a hypothetic anion pre-binding site between loop Cβ3-Cα1 and loop Cβ4-Cα2[43], forms an interface with TM12, TM13, and TM14 (Fig. 4d). Disease-associated mutations Y556C, F667C, and G672E of this region (Supplementary Fig. 5a), would significantly affect Cl⁻ and I⁻ transport[44].

STAS domains of SLC26 transporters have been hypothesized to serve as protein-protein interaction modules. The intracellular domain of Receptor tyrosine kinase Ephrin type-A receptor 2 (EphA2) is co-precipitated with pendrin in immunoprecipitation, and it can guide internalization with pendrin[45]. Besides, the GTPase activator and scaffold protein IQGAP-1 binds the STAS domain of hPDS/SLC26A4 and enhances its anion exchange activity[46]. When the protomers in different conformations were superimposed, the overall structure of the STAS domain had not much difference, except the helix Cα1b rotated about 6° from the inward state to the outward state (Fig. 3i, j). The movement of helix Cα1b might induce a larger conformation change of the missing IVS region that links to Cα1b (Fig. 4a). Interestingly, Cα1b is right below the anion transport pathway and provides a completely positively charged platform that is conserved among family members (Supplementary Fig. 6), which might help with the anion recruitment. Therefore, anion recruiting and/or intracellular protein signaling might induce conformation change of STAS domains, which in turn regulates the anion transport/exchange activities.

### Structural comparison of pendrin, prestin, and SLC26A9

The overall fold of pendrin is similar to prestin[23,28] and SLC26A9[21,22], however, the atomic cryo-EM structures of pendrin reveal its intrinsic features of asymmetric homodimer as an exchanger, given the symmetric homodimers resolved for all other SLC26 family members[21–23,28]. The structures of mouse SLC26A9 have two states: inward-open (PDB 6RTC) and intermediate (PDB 6RTF) states[21]. The dolphin prestin structures include several states, including an inward-like prestin-SO₄²⁻ (PDB 7S9C), intermediate-like prestin-Cl⁻ (PDB 7S8X) and several other states in between these two states[23]. When superimposing the inward-state protomers, the TMD regions of pendrin, prestin, and SLC26A9 are relatively similar (RMSD < 1.3 Å, Supplementary Table 2). The major

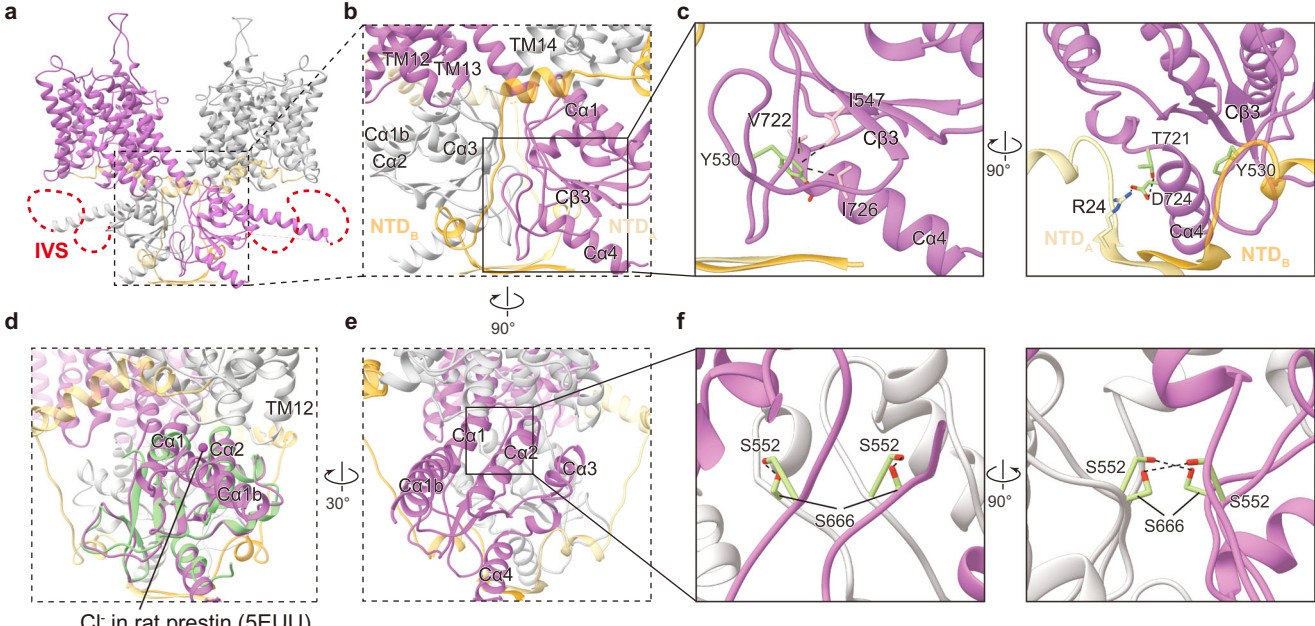

**Fig. 4 | STAS domain of pendrin. a** Structure model of pendrin-Cl. Protomer A is colored in orchid and orange, protomer B is colored in light grey and light yellow. **b** The dimerization interface around the STAS domain. **c** Details of STAS domain. Y530, T721, and D724 and their interactions are indicated by dot lines. **d** Comparison of STAS domain of pendrin-Cl with that of rat prestin (crystal

structure). Rat prestin-Cl crystal structure (PDB 5EUU) is colored light green. Cl⁻ in the hypothetic pre-binding site is pointed out. **e** A 90° rotated view of (**b**). **f** Details of the interface between two STAS domains. Hydrogen bonds between S552 and S666 are indicated.

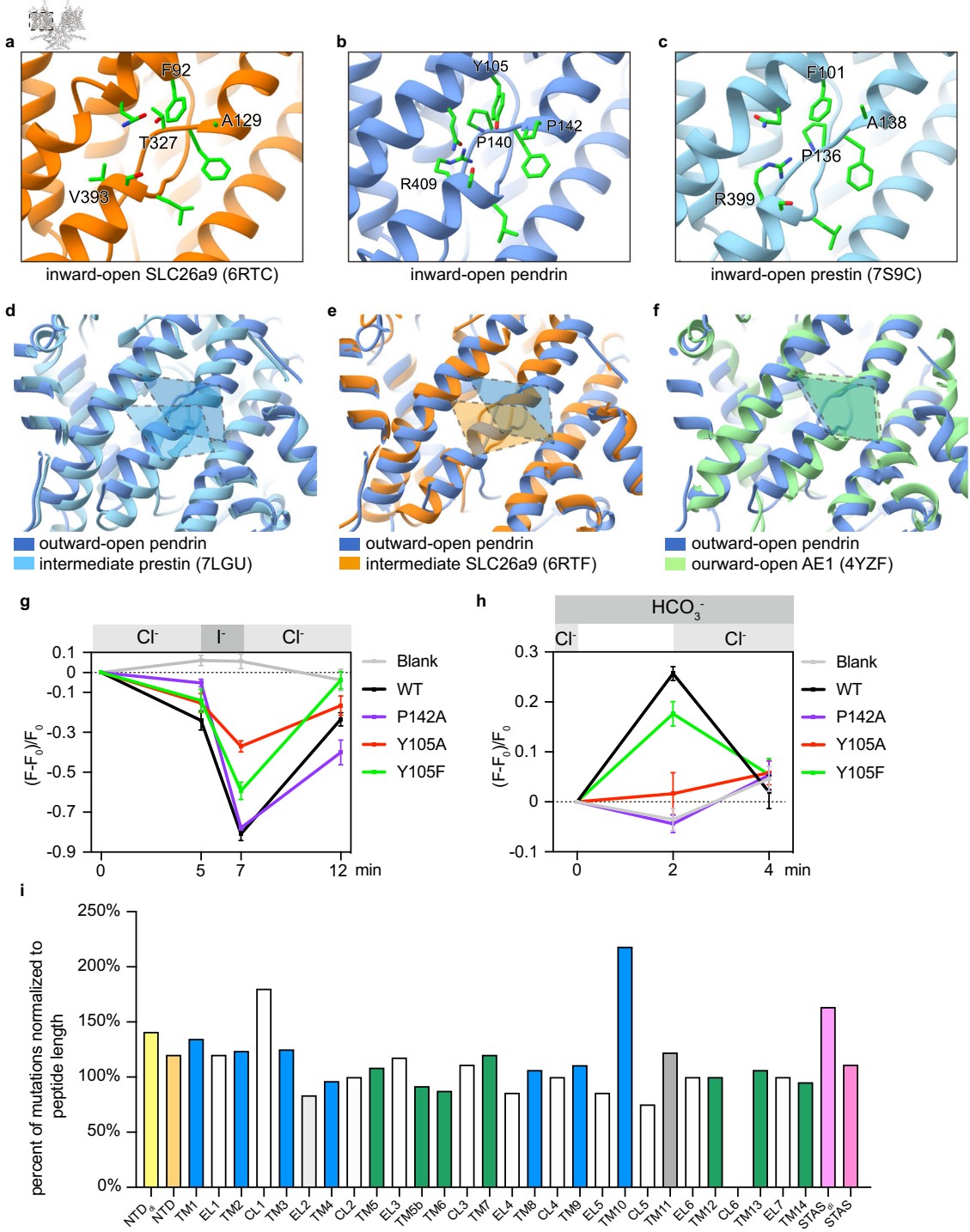

**Fig. 5 | Anion binding pocket of pendrin, prestin, and SLC26A9 and disease-associated variants of pendrin. a–c** Details of inward-open anion binding pocket of pendrin, prestin, and SLC26A9. The important residues are shown. **d–f** Comparison of anion binding pocket positions among outward-open pendrin, intermediate prestin, intermediate SLC26A9, and outward-open AE1. **g** Fluorescence intensity change in HEK293T cells in $Cl^-/I^-$ exchange assay. $n = 5$ cells examined over 3 independent experiments. Blank, EYFP-transfected cells. Data are presented as mean values, error bars indicate SD. **h** Fluorescence intensity change in HEK293T cells in $Cl^-/HCO_3^-$ exchange assay. $n = 5$ cells examined over 3 independent experiments. Blank, non-transfected cells. Data are presented as mean values, error bars indicate SD. **i** Location summary of 761 disease-associated mutations in the resolved 655 amino acids of our pendrin model. Disease-associated mutations are cited from the deafness variation database (https://deafnessvariationdatabase.org; accessed on the 20 Sep 2022). $NTD_{di}$ indicates the residue in the dimerization interface of NTD, EL indicates the extracellular loop, CL indicates the cytosolic loop, and $STAS_{di}$ indicates the residue in the dimerization interface of STAS.

difference is that the STAS domain of SLC26A9 has a distinct angular offset from pendrin and prestin, while the latter two overlap with each other (Supplementary Fig. 7c). Comparing outward-open pendrin with prestin and SLC26A9 in various intermediate states, we could observe

more differences. Among all SLC26 structures, the outward-open pendrin's core region has the largest rotation, and its anion binding pocket is also the uppermost, close to the outward-open state of AE1 (PDB 4YZF, Fig. 5d–f).

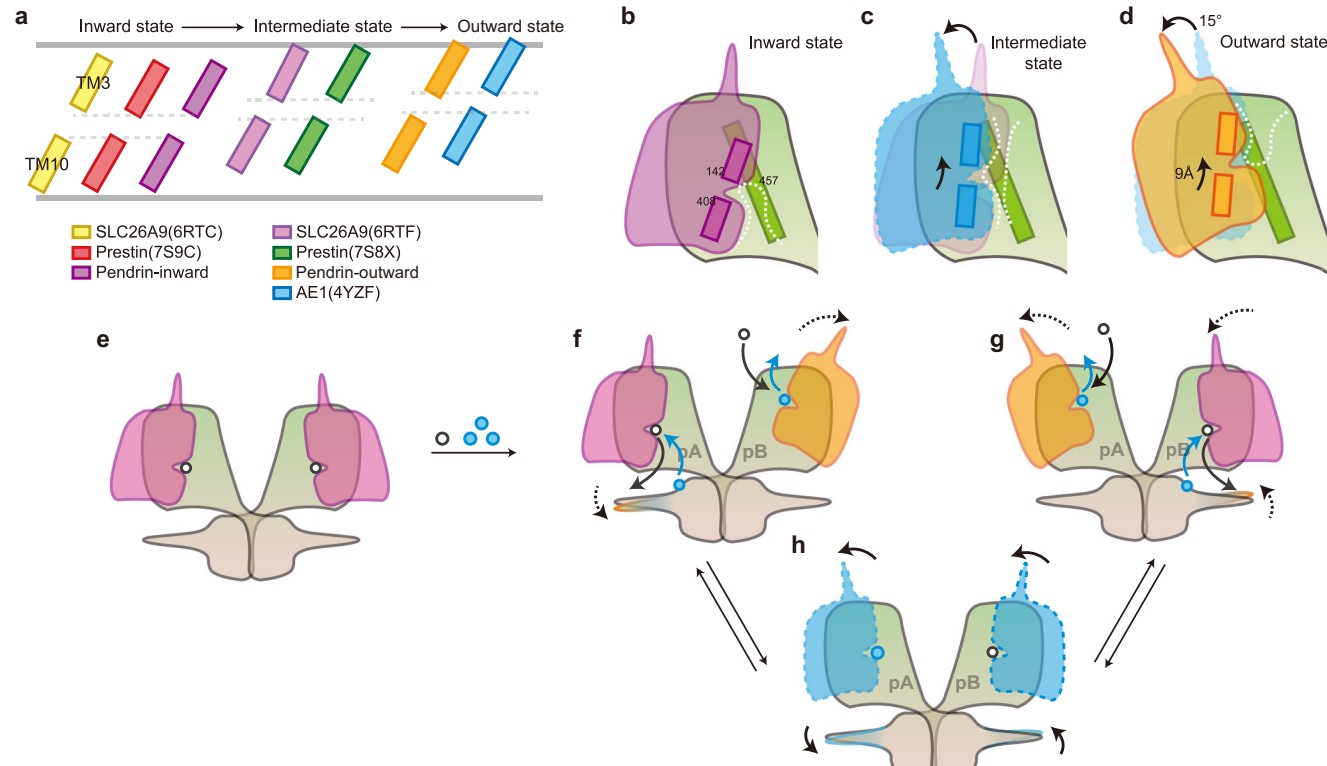

**Fig. 6 | Comparison among family members and exchange mechanism. a** TM3 and TM10 defined the pocket position of representative states of SLC members' pockets. **b–d** Schematic representation showing a complete alternate opening in TMD. **e–h** Schematic representation showing a hypothetic exchange mechanism.

**e** A thermodynamically stable symmetric inward homodimer in resting state. **f** binding of anions pair activates the transport of one protomer. Allosterism of two TMDs processes the inverted alternate-access mechanism. pA and pB: protomer A and protomer B, respectively.

When focused on anion binding pocket, within a conserved framework, residue differences were found at key sites between pendrin, prestin, and SLC26A9. Sequence alignment showed that pendrin residues Q101, F141, L407, and S408 are invariant, but not Y105, P140, P142, and R409 (Fig. 5a–c and Supplementary Fig. 8). Different from the allelic residue of prestin and SLC26A9's phenylalanine, pendrin Y105 has the phenolic hydroxyl group. So we performed the exchange assays to detect Y105F and Y105A. Y105F retained its Cl⁻/I⁻ exchange ability, although the transport activity of Cl⁻ and I⁻ decreased. Moreover, the Cl⁻/HCO₃⁻ exchange activity of Y105F significantly decreased (Fig. 5g, h). For Y105A, I⁻ transport function was further impaired, and Cl⁻/HCO₃⁻ exchange is loss of function (Fig. 5g, h and Supplementary Fig. 9, 10). These results proved that Y105 increases the charge of the binding pocket and enhances the electrostatic attraction to anions. Pendrin's P140-F141-P142 hydrophobic fragment has slightly different allelic residues in prestin and SLC26A9, PFA, and TFA, respectively. We measured the exchange function of P142A because P142A mutation would lose pi-pi stacking between F141 and P142. Notably, P142A remains Cl⁻/I⁻ exchange function, but loses Cl⁻/HCO₃⁻ exchange function (Fig. 5g, h). This result is consistent with the fact that SLC26A9 is permeable to Cl⁻ but not HCO₃⁻. The diversity of these three consecutively amino acids tunes the surface shape and charge of the pocket and may cause differences in interaction intensity between the pocket and anions. Finally, pendrin R409 is conserved in most SLC26A members except SLC26A1, SLC26A2, and SLC26A9. In SLC26A1 and SLC26A2 it is replaced by Lysine, which might be related to the specific function in SO₄²⁻ transport; while in SLC26A9, this residue was replaced by Valine. Within the binding pocket, pendrin R409 provides the only positive charge and forms multiple hydrogen bonds with neighboring residues for stability, as does in Prestin. However, Valine in SLC26A9 only has a hydrophobic short hydrophobic side chain, which weakens

the overall positive charge of the pocket. In summary, the surface charge of the binding pocket might directly define the anion selectivity, which may be the reason why these three members are so different in terms of anion selectivity and function.

## Mapping of clinical disease-associated mutations

Pendrin is a well-studied family member due to the enormous mutations that have been implicated in patients. The Deafness Variation Database [https://deafnessvariationdatabase.org/] lists over 8,000 sequence variants detected to date within the pendrin gene, of which 810 are missense mutations[29]. Our cryo-EM pendrin structures make it possible to rationalize these previous mutational studies. Of all the identified mutations, we analyzed 761 missense mutations mapped in our model (except the missing IVS and CT regions), and most remain to be examined experimentally (Supplementary Data 1).

Statistically, these mutations are distributed in almost every region except the CL6 loop of pendrin (Fig. 6i). First, CL1-TM3 and EL5-TM10, which build the characteristic anion-binding pocket, are the mutation hotspots and the mutations in these regions contribute enormously to pathogenicity (Supplementary Fig. 9 and 11a). In the intracellular vestibule entering the pocket along the anion exchange pathway, many mutations would change the local charge (V370E, S415R, G460E, D467V, R470C). Meanwhile, on the extracellular side, the anion-binding site also has some similar mutations (Q235R, K447E, D271H, D271G). These mutations could strongly interfere with anion affinity and directly disrupt anion recruitment. In the TMD cleft, most mutations happened to change the interface profile, which might cause a clash during the alternate opening and closing (core region: S93N, S93R, V97L, V97E, L100P, Q101P, A104T, A104V, L108P, V143L, A352T, S355P, V359E, A362D, E414K; gate region: V220F, A227P, V231M, V231L, V231E, V304E, S314L, V454L, N457Y). Additionally, in the

hydrophobic periphery of TMD, mutations mainly occur to change the hydrophobicity or surface profile (details are listed in Supplementary Data). These mutations affect the stability of pendrin in the cell membrane and may also disrupt protein-lipid interactions. As to the STAS domain, the dimerization interface has a very high mutation rate affecting dimerization, which eventually causes a high pathogenicity rate (Supplementary Fig. 9). Furthermore, the positively charged platform (formed by Cα1, Cα1b, and Cα2), the mutations with charge change could also cause dysfunction of STAS domain signaling (Y556H, Y556D, G557D, D560N, R576G, Q589E, K593E, G595E). Except for all the above-mentioned mutations, there are many mutations leading to steric hindrance inside individual rigid domains, which would cause misfolding and mis-localization of the whole protein. Although some mutations are not yet explained structurally, hereby we classify the previously reported disease-causing mutations from a structural point of view.

### The inverted alternate-access exchange mechanism

The competitive binding of the anion pair in the binding pocket determines the conformation of the TMD, and likely modulates that of STAS as well. Here, we have resolved a wide spectrum of pendrin structures flash frozen under various conditions. Based on structural analysis, we hypothesized the working model of pendrin as an anion exchanger. The symmetric inward-open conformation most likely represents the energetically favorite state, existing in the single binding-anion condition, $Cl^-$ for instance (Fig. 6b, e). When bicarbonate was added to a high concentration, $HCO_3^-$ replaces $Cl^-$ from the binding site in protomer B. This causes local conformation changes, with the binding pocket of protomer B translating towards the extracellular side, while protomer A stays unchanged (Fig. 6f and Supplementary Movie 1). To accommodate this change, the core region rotates about 15° against the gate region, a movement reminiscent of the elevator transport mechanism[21] (Fig. 6c). Eventually this conformational change would end up with the outward-open state (Fig. 6d), at which $HCO_3^-$ diffuses out and $Cl^-$ binds to reverse the exchange process (Fig. 6f). Considering the significant cross-section area increase from inward-open state to outward-open state, we hypothesize that when one protomer is outward-open, the other one would be inward-open to minimize their effects on the outer leaflet (Supplementary Fig 4). Therefore, the coincidence of secretion and uptake in this asymmetric homodimer shapes the molecular basis of the electroneutral exchange of pendrin with an inverted alternate-access mechanism.

Moreover, some important disease-associated variants were mapped on the structure, functional studies were also performed to interpret the structure-function relationships. All the above-mentioned studies would provide a framework for us to understand more disease-associated mutations and could shed light on therapeutic discovery.

## Methods

### Constructs and cell culture

DNA encoding full-length mouse pendrin (UniProt ID: Q9R155-1) was synthesized and subcloned into the pEZT BacMam vector, with an N-terminal 3X Flag tag and a 3C protease cleavage site. Primers: F: 5′-tttttttttttctgttccagggggcccatggcagcgcggggcggcag-3′; R: 5′-tgcagtcgcggccgctcagtgatggtgatggtgatgggaagcaagtctacgcatggcctcatcc-3′. For fluorescence assay, full-length mouse pendrin was shuttled into a pcDNA 3.1 vector and N-terminal tags were replaced by a mCherry or EGFP fluorescent protein sequence. Primers: F: 5′-gacgagctgtacaagatggcagcgcggggcggcag-3′; R: 5′-tctagactcgagcggccgctcagtgatggtgat-3′. Single point mutations were PCR amplified and subcloned based on wild-type plasmids. Primers: P142A-F: 5′-ttggcccttcgccgtggtcagt-3′; P142A-R: 5′-ccattaaactgaccacggcgaaaggg-3′; Y105A-F: 5′-atggctgctgccctgctggca-3′; Y105A-R: 5′-agggcagcagcagccatcccttgca-3′; Y105F-F: 5′-ggatggctttgccctgctggc-3′; Y105F-R: 5′-cagggcaaaagccatcccttgca-3′.

Sf9 cells were cultured in ESF921 medium at 27 °C and used for BacMam virus amplification. The P3 virus used for infection was precipitated by PEG 6000 and re-suspended with 1/10 volume of OPM medium. HEK293E cells used for protein expression and purification were cultured in OPM medium at 37 °C and 5% $CO_2$. Adherent HEK293T cells for fluorescence assay were cultured in DMEM medium supplemented with penicillin/streptomycin and 10% FBS. Commercial cell lines were obtained from the following vendors: Sf9 (express system, 94-001 F), HEK293-E (Procell Life Science&Technology, CL-0311), HEK293T (Thermo Fisher Scientific®, NC0260915).

### Protein expression and purification

HEK293E cells cultured at 37 °C with a density of ~2.3 × 10⁶ /ml were infected with resuspended BacMam virus at a volume ratio of 100:1. A final concentration of 10 mM sodium butyrate was added 12 h after infection, and cells were cultured at 37 °C for another 48 h. For fluorescence assays, adherent HEK293T cells were cultured in 6-well plates with a cell density of ~70% and transfected with plasmids-polyethylenimine (PEI 25 K, PolyscienceInc.) mix.

All subsequent purification procedures were carried out at 4 °C. Two liters of HEK293E cells were harvested and resuspended in buffer containing 20 mM Tris pH 8.0, 150 mM NaCl and protease inhibitors (0.8 mM aprotinin, 2 mg/ml leupeptin and 2 mM pepstatin A), and 1.5% (w/v) digitonin power was added into cell suspension and incubated for three hours. After centrifugation at 39,000 x *g* for an hour, the supernatant was filtered through a 0.45 μm filter and incubated with anti-flag affinity resin. The resin was washed with 20 mM Tris HCl pH 8.0, 150 mM NaCl, 5 Mm Mg-ATP, 0.02% (w/v) glycol-diosgenin (GDN) to remove heat shock proteins, and eluted with 0.3 mg/mL Flag peptide.

For pendrin-Cl, the concentrate was loaded onto a size-exclusion chromatography (SEC) column (Superose 6 Increase 10/300 GL) equilibrated with a buffer containing 20 mM Tris HCl pH 8.0, 150 mM NaCl, 0.02% GDN (Supplementary Fig. 1a). The peak fraction was analyzed by SDS-PAGE and concentrated to ~1.7 mg/mL for cryo-EM sample preparation. For pendrin-Cl/HCO₃ and pendrin-Cl/I, an SEC buffer containing 20 mM Tris pH 8.0, 100 mM NaCl, 0.02% GDN was used. And an equal volume of 300 mM NaHCO₃ (the final pH of sample pendrin-Cl/HCO₃ was 8.3) or NaI was added into peak fractions before concentrating. For pendrin pendrin-HCO₃ and pendrin-HCO₃/I, an SEC buffer containing 20 mM HEPES pH 8.0, 60 mM Na₂SO4, 0.02% GDN was used to replace $Cl^-$, and a final concentration of 10 mM NaHCO₃ was added into peak fractions during the concentration and the final pH of sample pendrin-HCO₃ was 8.3. Sample pendrin-HCO₃/I was added with a final concentration of 33 mM NaI based on sample pendrin-HCO₃.

### Cryo-EM grid preparation and data collection

Each pendrin sample was concentrated to about 1.7 mg/mL, and 3 μL of the protein was placed on glow-discharged grids (NiTi Au 400# R1.2/1.3), then the grid was blotted for 2.0 s with blotting force −2 and flash-frozen in liquid ethane with Vitrobot Mark IV (Thermo Fisher Scientific®).

Some pendrin cryo-EM datasets (including pendrin-Cl, pendrin-HCO₃, and pendrin-HCO₃/I) were collected on Titan Krios (Thermo Fisher Scientific) operated at 300 kV, equipped with K2 Summit direct electron detection device (Gatan) and BioQuantum energy filter (Gatan) set to a slit width of 20 eV. Automated data acquisition was carried out with SerialEM software[47] through the beam-image shift method[48]. Others (pendrin-Cl/HCO₃, pendrin-Cl/I) were collected on the same microscope with the camera upgraded to K3 Summit (Gatan®) using a slit width of 20 eV.

For the data collected with K2 camera, movies were taken in the super-resolution mode at a nominal magnification of X130,000, corresponding to a physical pixel size of 1.046 Å, and a defocus range

from −1.2 to −2.2 μm. Each movie stack was dose-fractionated to 36 frames with a total exposure dose of about 53 e$^-$/Å$^2$ and an exposure time of 7.2 s.

For data collected with K3 camera, movies were taken in the super-resolution mode at a nominal magnification of X81,000, corresponding to a physical pixel size of 1.064 Å, and a defocus range from −1.2 to −2.2 μm. Each movie stack was dose-fractionated to 40 frames with a total exposure dose of about 58 e$^-$/Å$^2$ and an exposure time of 3.0 s.

Details are listed in Supplementary Table 3.

## Cryo-EM image processing

The routine processing of all datasets was carried out with the same procedure. Movie stacks were binned 2 × 2, dose weighted, and motion-corrected using MotionCor2[49] within RELION[50]. Parameters of the contrast transfer function (CTF) were estimated by using Gctf[51]. Bad images were excluded upon ice condition, defocus range, and estimated resolution. The remaining good images were imported into cryoSPARC[52] for further patched CTF-estimating, blob-picking, and 2D classification. Several good 2D classes were selected as the template for template-picking. From the 2D classification classes, good particles from blob-picking and template-picking were merged and deduplicated. Two rounds of 3D classification were done in RELION, using an initial model generated by cryoSPARC as a reference. The particles from the high-quality classes were selected and re-extracted without binning. Auto-refinement was performed with C1 symmetry, with further CTF-refinement and particle polishing, yielding consensus maps at the range of 3.4−3.8 Å. Then the well-aligned particles were imported back into cryoSPARC for 3D variability analysis (3DVA) regarding the conformational change. Surprisingly, these samples containing two anions all had different conformations including inward-open state and outward state. Preliminary data processing was finished so far, and variant strategies were employed to classify different conformations. To further investigate possible anion densities in the binding pocket of the outward-open state, we did local refinement of outward-open protomer from pendrin-Cl/HCO$_3$ io data.

The reported resolutions are all based on the gold-standard Fourier shell correlation (FSC) 0.143 criterion. All the visualization and evaluation of 3D density maps were performed with UCSF Chimera[53]. The above procedures of data processing are summarized in Supplementary Fig. 12. These sharpened maps were generated by DeepEMhancer[54] and then "vop zflip" to get the correct handedness in UCSF Chimera for subsequent model building and structural analysis.

## Model building and structure refinement

Model building of inward-open state with C2 symmetry was refined from the predicted model in Alpha Fold with Coot[55] based on the high-resolution 3.3 Å pendrin-Cl cryo-EM map. Most residues were clearly resolved in our cryo-EM map, and total 665 (18-737Δ596-650) amino acid residues were constructed for each monomer. N-terminal 1-17, IVS 596-650, and C-terminal 738-780 were not modeled because the corresponding density was absent in the map. Structure refinement was performed with PHENIX[56] with secondary structure and geometry restraints to prevent structure overfitting. Model building of the outward state with C2 symmetry was refined based on the 3.6 Å pendrin-Cl/HCO$_3$ cryo-EM map with the same workflow above. Other C1 symmetrical models were refined from C2 models based on their own maps. Statistics associated with data collection, 3D reconstruction, and model refinement can be found in Supplementary Table 3.

## Fluorescence anion exchange assay

Fluorescence assays were taken 48 h after transfection. Measurements of intracellular pH in HEK 293 T cells transiently transfected with mCherry-pendrin plasmids were performed using a pH-sensitive fluorescent probe BCECF (2′,7′-bis-(2-carboxyethyl)−5-(and-6)-carboxyfluorescein). After dye loading (15 min, 37 °C), the cells were perfused with a buffer containing 110 mM NaCl, 25 mM NaHCO$_3$, 10 mM glucose, and 5 mM HEPES pH 7.5, and BCECF fluorescence was recorded at the excitation wavelengths of 488 nm and the emission wavelengths of 530 ± 10 nm by confocal microscope (Leica TCS SP8) at 37 °C with a 5% CO2 gassing. Then cells were treated alternately by Cl$^-$free buffer (NaCl is replaced by sodium gluconate) and Cl$^-$containing buffer and photographed under the same conditions. Cl$^-$/HCO$_3^-$ exchange activities were estimated from fluorescence intensity change.

For in vivo Cl$^-$/I$^-$ measurements, HEK 293 T was transiently co-transfected with EYFP and mCherry-pendrin plasmids. After titration, we selected the Cl$^-$/I$^-$ concentration tuning the fluorescence intensity within the appropriate range of fluorescence microscopy. Cells were treated alternately by Cl$^-$containing buffer (140 mM NaCl, 10 mM glucose, and 5 mM HEPES pH 7.5) and I$^-$containing buffer (25 mM NaI, 115 mM sodium gluconate, 10 mM glucose, and 5 mM HEPES pH 7.5). The EYFP fluorescence was recorded at the excitation wavelengths of 488 nm and the emission wavelengths of 530 ± 10 nm by the same confocal microscope at 37 °C with a 5% CO2 gassing. Cl$^-$/I$^-$ exchange activities were estimated from fluorescence intensity change[57].

The fluorescence intensity was measured by ImageJ and calculated by GraphPad Prism 8.

## Reporting summary

Further information on research design is available in the Nature Portfolio Reporting Summary linked to this article.

## Data availability

The data that support this study are available from the corresponding authors upon request. The cryo-EM density maps have been deposited in the Electron Microscopy Data Bank (EMDB) under accession codes EMD-32555 (pendrin-Cl), EMD-32561 (pendrin-HCO3), EMD-32577 (pendrin-Cl/I$_{ii}$), EMD-32580 (pendrin-Cl-I$_{io}$), EMD-32576 (pendrin-Cl/HCO3$_{ii}$), EMD-32578 (pendrin-Cl/HCO3), EMD-32583 (pendrin-Cl/HCO3$_{oo}$), EMD-32574 (pendrin-HCO3/I$_{ii}$), EMD-32579 (pendrin-HCO3/I$_{io}$). The local refinement map of pendrin-Cl/HCO3 io has deposited under accession code EMD-33232 code 33232. Coordinates have been deposited in the Protein Data bank (PDB) under accession codes 7WK1 (pendin-Cl), 7WK7 (pendrin-HCO3), 7WL8 (pendrin-Cl/I$_{ii}$), 7WL7 (pendrin-Cl/HCO3$_{ii}$), 7WL9 (pendrin-Cl/HCO3 $_{io}$), 7WLE (pendrin-Cl/HCO3 $_{oo}$), 7WL2 (pendrin-HCO3/I $_{ii}$), 7WLA (pendrin-HCO3/I $_{io}$). Previously published PDB entries cited here can be accessed under accession codes 7S9C, 6RTC, 7LGU, 6RTF, and 4YZF. Source data are supplied as a Source Data file. Source data are provided with this paper.

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

## Acknowledgements

We thank the Center of Cryo-Electron Microscopy, Fudan University for the support on cryo-EM data collection. This work was supported by the National Natural Science Foundation of China (31970146 to ZC), the Ministry of Science and Technology of China (2021YFC2302500 to LS), the National Key Research and Development Program of China grant (2020YFA0908201 to YS), National Natural Science Foundation of China grant (82171148 to YS), Science and Technology Commission of Shanghai Municipality (21S11905100 to YS) and Guangzhou Laboratory (SRPG22-003 to LS).

## Author contributions

Z.C., L.S., and Y.S. conceived the project; Q.L., Q.M. cloned, and Q.L. expressed, and purified proteins; X.Z., Z.C., and Q.L. collected and processed cryo-EM data. Q.L. and L.S. built the atomic models; Q.L., H.H., and W.Z. performed fluorescence assays; F.W., Y.C., A.H., W.Z., Q.M., Y.H., L.H., Y.S., M.Z., Z.L., and G.L. were involved to analyze and discuss the data; Q.L., L.S., and Z.C. wrote the manuscript and all the authors reviewed it.

## Competing interests

The authors declare no competing interests.
