## [Peer Review File · Nature Communications]

Asymmetric Pendrin Homodimer Reveals its Molecular Mechanism as Anion ExchangerReviewers' Comments:

Reviewer #1:

Remarks to the Author:

Pendrin is located in epithelial cells of inner ear, functioning in HCO₃⁻ secretion into endolymph. Dysfunction of pendrin can result in hearing loss. This work performed structural and functional study on pendrin. The authors solved several cryo-EM structures of pendrin, including an asymmetric structure with different conformations solved in a dimer structure. The pendrin structures captured in this work provide valuable insight into conformational changes in the transport cycle of pendrin and other SLC26 family proteins. This work also revealed the substrate binding site for pendrin and investigated the mutations of pendrin. There are some points that need to be addressed in the manuscript.

Major points:

1. Pendrin exists in a symmetric inward-open dimer conformation under single substrate condition, Cl⁻ or HCO₃⁻. Addition of second substrate can induce the asymmetric dimer conformation of pendrin. It seems that there should be an interaction mechanism between the two protomers within a dimer molecule. How does the conformational change of one protomer influence the conformation of the other? Is there a possible conformational transduction pathway within pendrin asymmetric dimer? The authors should discuss these points.
2. Can the asymmetric dimer conformation be captured in the condition of Cl⁻ alone or HCO₃⁻ alone? It is recommended to re-classify the dataset using the asymmetric dimer conformation as one of references.
3. line 179-180, "Pendrin purified in 100 mM NaCl was mixed with the same volume of 300 mM NaHCO₃". How long did it take to create the mixture? How long was the incubation time? Is the sequence of incubation buffers critical? What results would be obtained if the authors used the protein samples incubated in a buffer containing 50 mM NaCl and 150 mM NaHCO₃ to solve the cryo-EM structure?
4. Cholesterol bound in pendrin needs more validation. The density between the TM domains is not necessary to be the two cholesterols. One cannot rule out the artifact induced by C2 symmetry application during data processing.
5. The electrophysiology results of P142 are not consistent in Fig 2e and Extended data Fig. 1d. How to explain the inconsistency?
6. Y105 and S408 seem to be the dominant residues in Cl⁻ coordination, but the electrophysiology results of Y105A and S408A are not significant different (Extended data Fig. 1) or only mildly significant (Fig 2e). At the same time, N457 from gate domain shows the most significant variation in electrophysiology assay. Obviously N457 is not involved in Cl⁻ coordination directly. Therefore, electrophysiology results so far are rather confusing. Besides, mutations to alanine may not always be suitable, since mutations like Y105A and P142A are making huge difference to the original residues, which could induce destruction to protein behavior, just like the authors suggest in the case of P142R and P140H. This reviewer suggests to check on the protein behavior before electrophysiology assay.
7. Electrophysiology assays are needed to confirmed function of S408 and L407 in HCO₃⁻ transport.
8. Pendrin was purified in 100 mM NaCl buffer first and then 300 mM HCO₃⁻ is added before protein concentration for cryo-EM samples. Cl⁻ in the protein could be replaced by HCO₃⁻. The binding affinity between pendrin to Cl⁻ and HCO₃⁻ need to be measured in the condition of cryo-EM sample buffer.
9. The transport path of substrates in pendrin should be analyzed by HOLE.
10. Is there any direct interaction between Ca1b and transmembrane domain? How is its rotation related to the TMD conformational change?

Minor points:

11. line99, No robust data in Extended data fig 1 supports the statement "pendrin forms a dimer as other family member". The only related data is g panel, but the size exclusion chromatography is not enough to suggest "Pendrin forms a dimer".
12. line175, "... pendrin is exposed a mixture ..." should be "pendrin is exposed to a mixture ..."

13. line 237, "despite the completeness differences". This sentence is confusing.
14. line244, "The starting residues 515-545 of STAS domain form a long loop region, however, it is well resolved due to the interactions with C β 3, Ca5 and NTD". It is recommended to remove the word "However" from this sentence.
15. line280, "significantly contribute to the initiation of anion transport within STAS domain". There is no robust support for anion transport initiating from STAS domain. Citation should add if any paper supports this opinion.
16. Fig 1b is not cited in the paper. Besides, topology should contain the information of cytosolic and extracellular side. It would be perfect to add this in the panel. The mapped mutations should cite the original papers.
17. Fig 4f, S552 and S666 should be labeled more clearly.

Reviewer #2:

Remarks to the Author:

SLC26A4 encodes an electroneutral anion exchanger, pendrin, and a large number of genetic variants associated with syndromic (Pendred syndrome, PDS) or nonsyndromic (DFNB4) hearing loss have been identified. Structural information of the pendrin protein is expected to greatly facilitate efforts in defining the pathological roles of these SLC26A4 variants. This study by Liu et al. reports cryo-EM structures of homodimeric mouse pendrin obtained under various anion substrate conditions. Overall, these pendrin structures look very similar to those recently reported by others for SLC26A5 (prestin) and SLC26A9. However, this study is novel in that an outward-open state was captured for the first time for SLC26 family of proteins. It is exciting that this outward-open structure was found in an asymmetric pendrin homodimer (outward-open + inward-open), which would provide mechanistic insights as to how electroneutral 1:1 anion exchange is mediated by the pendrin protein. I am enthusiastic about the cryo-EM pendrin structures reported in this paper; however, the manuscript contains multiple issues, especially in their functional assays, that need to be addressed. Please see below for my specific comments.

Major points

(1) Figs. 2e, 4d, and Extended Data Fig. 1. The authors tried to assess the anion transport functions of pendrin and its mutants by whole-cell voltage clamp. What is the rationale for using such an electrophysiological method for measuring pendrin's transport activity? It is my understanding that pendrin exchanges bicarbonate or iodide with chloride in a 1:1 electroneutral manner. Chloride/chloride exchange is also possible, but none of these exchanges would result in net charge movement (electroneutral). Why were the BCECF- and YFP-based optical methods (Fig. 5) not used throughout the manuscript?

(2) Fluorometric iodide/chloride antiport assay (Fig. 5f). I guess that fluorescence data were collected at a higher temporal resolution and Fig. 5f shows a summary of corrected fluorescence data only for four time points. Please show examples of raw fluorescence traces as well. Please also mention how the fluorescence data were corrected and what the error bars indicate (SD or SE). In this assay, it is important to confirm steady basal fluorescence ($\Delta F/\Delta t \sim 0$) before solution exchange and to determine the slope of fluorescence change ($\Delta F\%/\Delta t$). Note that some pendrin constructs (WT, G209V, and G672E) show large reductions in basal fluorescence even before solution exchange, and that the direction of fluorescence change after solution exchange is opposite to what is expected – increase in fluorescence upon iodide perfusion for F209V, G672E; no fluorescence recovery after perfusing back to a chloride solution for G209V, G672E, and E303Q. I am not convinced that this assay was done properly. Also, statistics needs to be provided for any comparisons.

(3) Fluorometric bicarbonate/chloride antiport assay (Fig. 5g). A dual-excitation ratiometric pH-sensitive fluorescence probe, BCECF, was used; however, the authors monitored the fluorescence excited only with single excitation wavelength (488 nm). Consequently, the intracellular pH could not

be determined. This is problematic, because the response of BCECF to pH is not linear, and because it is likely that the initial intracellular pH before solution exchange differs among the pendrin constructs. Thus, the data cannot be trusted in the present form. If a shorter wavelength light source were not available or if switching of excitation wavelengths were cumbersome, I would suggest to use a dual-emission ratiometric pH-sensitive fluorescence probe (e.g., SNARF, PMID: 31599023). Please show examples of intracellular pH traces (before/after solution exchange) along with a summary of transport activities ($[H^+]/\text{sec}$). Again, statistics needs to be provided.

(4) The authors should take advantage of the large number of PDS/DFNB4-associated SLC26A4 variants identified in patients to date and previous functional studies. For example, the presence of multiple disease-associated variants at P142, S408, and N457 (i.e., P142R, P142L, P142S, S408Y, S408P, S408F, N457K, N457D, N457Y, and N457I) implies that these missense variants are likely pathogenic, and previous functional studies demonstrated that P142R, P142L, S408F, and N457K indeed impair the function of pendrin (PMIDs: 18310264; 26752218; 31599023). Now, the pendrin structure reported in this study accounts for the underlying pathological mechanisms for these variants. I do not think that it is crucial to collect additional functional data for P142A, S408A, and N457A (Figs. 2e, 4d, and Extended Data Fig. 1) to claim the functional importance of these residues. In my opinion, what is more important is to facilitate efforts in predicting the potential pathological roles of variants with uncertain significance (VUS). Currently, ~800 SLC26A4 missense variants have been identified, approximately half of which are VUS (<https://deafnessvariationdatabase.org>). I strongly encourage the authors to expand Extended data Table 3 by providing structure-based pathogenicity predictions for many more SLC26A4 missense variants rather than mostly duplicating previous experimental efforts (note that almost all deafness-associated SLC26A4 variants examined in Figs. 4 and 5 have already been characterized in previous studies).

(5) Fig. 3. How is the inward-open state found in the inward/outward asymmetric pendrin homodimer compared to those found in the inward-inward symmetric homodimer? Likewise, how is the outward-open state found in the inward/outward structure compared to those found in the outward-outward homodimer? Please describe if there are any noteworthy differences at any regions of the pendrin protein (e.g., N-terminus, dimerization interface, STAS, locations of the bound chloride and bicarbonate, etc.), as such information may provide significant mechanistic insights as to how conformational changes of the two protomers are coordinated to achieve 1:1 electroneutral antiport.

(6) Fig. 3b. Is it possible that just one bicarbonate ion binds to the outward-open state but with two distinct modes (e.g., one tightly bound and the other loosely bound)? Please describe in more detail.

(7) Extend data Fig. 3. How are anions bound to these structures? How are the inward- and outward-open structures compared to those found in other homodimeric structures (chloride/chloride, bicarbonate/chloride)? Extended data Table 1 indicates that there are differences, but readers would like to know where in the structures the differences are found.

(8) Throughout the manuscript, avoid using the terms "pathogenic mutation(s)". Instead, use "disease-associated variant(s)" or "deafness-associated variant(s)".

(9) The manuscript contains many typos and grammatical errors that need to be fixed.

Minor points

(1) Line 98. "no other oligomers was observed". I find a large elution shoulder preceding the pendrin peak in the SEC profile shown in Extended data Fig. 1g and smear above the pendrin band in Extended data Fig. 1h. What are they?

(2) Line 123. "view of g showing ~" should be "view of c showing ~".

(3) Figs. 2e and 4d and Extended data Fig. 1. The current data should be corrected for the cell

membrane capacitance, as larger cells tend to express larger amounts of pendrin. Also, it is problematic that different statistical conclusions are derived for Y105A, P142A and S408F in Fig. 2e vs. Extended data Fig. 1d. In any case, I don't think that this electrophysiological assay is appropriate for measuring pendrin's transport activity. These data should be removed from the manuscript (see above).

(4) Lines 147 and 158. Avoid saying "supposed to", as it sounds highly subjective.

(5) Line 148. "~ will result ~" should be changed to "likely cause".

(6) Lines 158-160. "~. Coincidentally, the allelic residue of pendrin P142 is Alanine in prestin and SLC26A9". However, nonmammalian prestin orthologs and SLC26A9 have chloride transport activities, which is inconsistent with the authors' argument here.

(7) Lines 236-240. "In addition, the density of relatively stable cholesterol can be seen in all conformations, despite the completeness differences. Since cholesterol is believed to influence the localization and diffusion of prestin in plasma membrane, this may be characteristic for the interactions between SLC26 family members and plasma membranes". What does "all conformations" mean? What is "completeness differences"? These sentences do not make sense.

(8) Line 276. Please show the locations of Y556, F667, and G672 in Fig. 4 (or in a supplementary figure).

(9) Lines 278-281. Is the "positively charged platform" conserved among the SLC26 family? Please mention.

(10) Lines 297-303. "~ the side chain of leucine may disrupt protein-lipid interactions, ~". The amino acid equivalent to F335 in pendrin is L325 in prestin. A lipid is found in close vicinity of L325 in the prestin structure (PDB: 7LGU), opposing the authors' speculation.

(11) Lines 308-309. "~ side chain of valine would increase steric hindrance of core-gate interface and contribute the positive surface charge". How could valine contribute to positive surface charge?

(12) Line 311. Please define "pre-binding site" or cite a reference.

(13) Lines 312-313. "In Cl-/I- exchange assays, G672E lost I- transport capacity, but maintains Cl- permeability (Fig. 5f,j)". The quality of the transport activity data is too low to determine the functional consequence of G672E. Also, chloride transport data are not provided for G672E.

(14) Lines 336-338. Intracellular retention of the pendrin protein can be seen even for wild-type as evident in Figs. 5h and 5m. Thus, it is ambiguous what the authors mean by "with a well-defined cellular localization".

(15) Lines 345-346. F335L may not affect lipid binding. See above.

(16) Lines 356-357. "For R185T at the core-gate interface and L236P at the protein-lipids interface, the charge change may lead to local misfolding". How could L236P affect charge?

(17) "The major difference is that the STAS domain of SLC26A9 has a distinct angular offset from pendrin and prestin, while the latter two basically overlap with each other". Please explain this difference graphically (in a supplementary figure).

(18) Line 57 in Extended Data Fig. 6. "UniProt", not "UniPort".

REVIEWER COMMENTS

Reviewer #1 (Remarks to the Author):

Pendrin is located in epithelial cells of inner ear, functioning in HCO_3^- secretion into endolymph. Dysfunction of pendrin can result in hearing loss. This work performed structural and functional study on pendrin. The authors solved several cryo-EM structures of pendrin, including an asymmetric structure with different conformations solved in a dimer structure. The pendrin structures captured in this work provide valuable insight into conformation changes in the transport cycle of pendrin and other SLC26 family proteins. This work also revealed the substrate binding site for pendrin and investigated the mutations of pendrin. There are some points that need to be addressed in the manuscript.

We appreciate the reviewer's efforts of reviewing our manuscript, providing so many constructive questions and suggestions and giving positive comments. We have answered the points one by one in the following, for reviewer and editor's further reviewing.

Major points:

1. Pendrin exists in a symmetric inward-open dimer conformation under single substrate condition, Cl^- or HCO_3^- . Addition of second substrate can induce the asymmetric dimer conformation of pendrin. It seems that there should be an interaction mechanism between the two protomers within a dimer molecule. How does the conformation change of one protomer influence the conformation of the other? Is there a possible conformational transduction pathway within pendrin asymmetric dimer? The authors should discuss these points.

This is a very good point. We are thinking about this question all the time: how these two protomers coordinate with each other? Here are several speculations:

1. In pendrin dimer, STAS domains form a dimeric knob and provide the major dimerization interface. Thus, at the beginning, we expected that the asymmetric dimer conformation might be regulated by STAS domain through its conformation change around the dimerization interface, which then induce two protomers cooperating with each other (Response Fig. 1a, transduction pathway marker by dark blue arrows). However, comparison of outward-open and inward-open pendrins showed that STAS domains are very similar. The dimerization interface is almost identical. The only difference is the rotation ($< 6^\circ$) of helix $\text{Ca}1\text{b}$, located at the periphery of the STAS domain (Fig. 3h). It seems that the slight conformation change of helix $\text{Ca}1\text{b}$ could not regulate the inward-outward transition of Pendrin.
2. We then investigated the influence of TMD conformation change on the lipid bilayer. When pendrin's one protomer transits from inward-open state to outward-open state, the cross-section area of the TMD in the external membrane leaflet becomes $\sim 27\%$ larger, and the cross-section area of the TMD in the internal membrane leaflet becomes $\sim 11\%$ smaller (Response Fig. 1b). Since pendrin is embedded in the cell membrane, the change of the cross-section area in both leaflets would cause stretch or compress to the lipids that are interacting with the protein surround. Therefore, we hypothesized that two protomers would incorporate the inverted alternate-accessing mechanism. In this way, one protomer's cross-section change

would be completely compensated by the other protomer. Then, this asymmetric homodimer is more physiologically favored than the symmetric inward-inward or outward-outward conformations. Therefore, we speculate the cell membrane tension provides the transduction pathway for two protomers to interact with each other (Response Fig. 1a, transduction pathway marker by red arrows).

We supplied the cross-section area figure of TMD in Supplementary Fig. 4. The discussion has been included in the “Inverted alternate-access exchange mechanism” section in the revised manuscript.

Response Fig. 1. a, Structure model of pendrin asymmetric structure. **b**, Supplementary Fig. 4b, At the outer leaflet and the inner leaflet, cross-section areas of pendrin in three conformations.

2. Can the asymmetric dimer conformation be captured in the condition of Cl⁻ alone or HCO₃⁻ alone? It is recommended to re-classify the dataset using the asymmetric dimer conformation as one of references.

Thanks for this suggestion. We tried to do the 3D classification using the asymmetric dimer conformation as the reference on both Cl⁻ alone and HCO₃⁻ alone data, but we didn't find any asymmetric class. We also did 3D-VA (a map video showing 3D variation of all pendrin particles) investigation, in the video, the map only showed symmetric inward-inward conformation. Therefore, the addition of the second kind of anions is the key point to induce the asymmetric conformation in vitro.

3. Line 179-180, “Pendrin purified in 100 mM NaCl was mixed with the same volume of 300 mM NaHCO₃”. How long did it take to create the mixture? How long was the incubation time? Is the sequence of incubation buffers critical? What results would be obtained if the authors used the protein samples incubated in a buffer containing 50 mM NaCl and 150 mM NaHCO₃ to solve the cryo-EM structure?

Thanks for this question. Here are the details of cryo-EM sample preparation: NaHCO₃ buffer was freshly prepared right before using it. Then the purified pendrin from gel filtration was mixed with fresh NaHCO₃ buffer, and concentrated immediately. The concentration and final high-speed centrifugation took about 20 mins, followed by 30 mins' cryo specimen preparation by Vitrobot mark IV. Therefore, the incubation time is about one hour (from the addition of the second anion to cryo-grid preparation).

As for the sequence of incubation buffer (the sequence of anion addition), we did not purify pendrin in HCO_3^- buffer first then add Cl^- because HCO_3^- is unstable in solution. In details, pH of freshly prepared 0.1 molar sodium bicarbonate solution is 8.3, and pH would get higher with higher concentration of sodium bicarbonate (PUBCHEM: CID 516892). And for long-term expose to air, bicarbonate will decompose into carbonate and carbon dioxide, and the solution will become alkaline (PUBCHEM: CID 10340). Unstable buffer pH would affect the quality of protein, so we did not try to purify pendrin with buffer containing high concentration of bicarbonate de novo.

We did try different sequences when incubating I⁻ with Cl⁻. Firstly, we successfully obtain pendrin-Cl/I (“Cl⁻ first, then I⁻”) specimen. Then we tried to purify pendrin in “firstly I⁻, then Cl⁻”. Unfortunately, in the buffer containing 100 mM NaI (20 mM Tris, 100 mM NaI, 0.02% GDN, pH 8.0), the protein aggregated severely during purification (Response Fig. 2), which was not suitable for structural study. Thus, neither bicarbonate nor iodide solution could purify pendrin de novo in the experiment.

Response Fig. 2. Size exclusion chromatography of pendrin purified in buffer containing NaCl and buffer containing NaI.

Even though we could not obtain evidence for different sequences of mixing two anions from the experiment directly, we speculate the appearance of asymmetric conformation does not rely on the sequence of adding anions, since we observed similar conformations in both $\text{Cl}^-/\text{HCO}_3^-$ and Cl^-/I^- buffers. Co-existence of two different transportable anions in the buffer triggered conformation transition of pendrin.

To response to “What results would be obtained if the authors used the protein samples incubated in a buffer containing 50 mM NaCl and 150 mM NaHCO_3 to solve the cryo-EM structure”, we did not purify protein in a buffer containing 50 mM NaCl and 150 mM NaHCO_3 from the beginning because 150 mM NaHCO_3 is not stable enough to survive the long-period protein purification.

4. Cholesterol bound in pendrin needs more validation. The density between the TM domains is not necessary to be the two cholesterol. One cannot rule out the artifact induced by C2 symmetry application during data processing.

Thank you for pointing out this. We checked the asymmetrical maps and observed similar lipid density in the same position (Response Fig. 3). However, indeed, we could not provide evidence to prove these densities are for cholesterol. Thus, we removed the cholesterol molecule from the model and the discussion about cholesterol.

**Response Fig. 3.** Lipid densities between two TMD in pendrin-Cl, pendrin-Cl/$HCO_3$ io and pendrin-$HCO_3$/I io.

5. The electrophysiology results of P142 are not consistent in Fig 2e and Extended data Fig. 1d. How to explain the inconsistency?

Thanks for this question. Our electrophysiological assay showed different results at positive 100 and negative 100 voltages. When performing this assay, we referred to the article about mouse SLC26A9 structure study (PMID: 31339488). In the electrophysiological assay of mouse SLC26A9, a trend difference between -100 and 100 was also observed.
To confirm whether the current we measured is a chloride current, we used a broad-spectrum chloride channel inhibitor NPPB to measure the current change. NPPB is known as a blocker of chloride transport in a variety of different cells. NPPB can inhibit the current of pendrin on cells. It was reported that 0.1 mM NPPB inhibited the pendrin-dependent iodide influx by 33% (PMID: 27161422) and reduced the pendrin-chloride uptake by 59.3±8.4% (PMID: 16791000). Unfortunately, the addition of inhibitors did not significantly suppress the current, indicating that the current we measured were mainly generated by other ions instead of chloride. Therefore, we removed the electrophysiological assay data from the manuscript, as the other reviewer suggested. As an exchanger, the anion exchange might not result in net charge movement.
YFP and pH probes are both widely used to detect the function of pendrin and its mutants. So, we re-examined the residues near the binding pocket using fluorescence experiments. As a result, the $Cl^-/I^-$ exchange of P142A still worked (Response Fig. 4a), but the ability of $Cl^-/HCO_3^-$ exchange was lost (Response Fig. 4b), which coincided with the results of the pathogenetic mutation P142R, and also with the fact that SLC26A9 could not transport $HCO_3^-$. We added these new results in main Fig. 5 and Supplementary Fig. 9 and 10.

Response Fig. 4. a, Supplementary Fig. 9b Curves of YFP fluorescence change, error bars indicate SD. **b**, Supplementary Fig. 10b Curves of BCECF fluorescence change, error bars indicate SD.

6. Y105 and S408 seem to be the dominant residues in Cl⁻ coordination, but the electrophysiology results of Y105A and S408A are not significantly different (Extended data Fig. 1) or only mildly significant (Fig 2e). At the same time, N457 from gate domain shows the most significant variation in electrophysiology assay. Obviously N457 is not involved in Cl⁻ coordination directly. Therefore, electrophysiology results so far are rather confusing. Besides, mutations to alanine may not always be suitable, since mutations like Y105A and P142A are making huge difference to the original residues, which could induce destruction to protein behavior, just like the authors suggest in the case of P142R and P140H. This reviewer suggests to check on the protein behavior before electrophysiology assay.

Thanks for pointing this out. As mentioned in major question 5, electrophysiology assay did not reflect the function of these mutations properly. So, we removed the electrophysiology assay data. Instead, we examined P142A, Y105A, Y105F, S408A and N457A using fluorescence experiments (Response Fig. 4). As for Y105, other than Y105A, we also tested Y105F mutant (in prestin and SLC26A9, allelic residue of Y105 is phenylalanine). As shown in exchange assays, Y105F partly reduced the anion exchange ability, while Y105A had severe reduction of exchange function. N457A did not affect exchange ability very much. And as for P142, it has been reported that pathogenetic mutation P142R abolished the Cl⁻/HCO₃⁻ exchange activity and pathogenetic mutation P142L reduced Cl⁻/I⁻ exchange. Therefore, we did not repeat these mutations. In addition, in prestin and SLC26a9, allelic residue of P142 is alanine, so we tested P142A. The result brought some information of difference between family members like we discussed in major question 5.

7. Electrophysiology assays are needed to confirm function of S408 and L407 in HCO₃⁻ transport.

Thanks for this suggestion. Pendrin is capable of Cl⁻ and I⁻ transport, but not HCO₃⁻ alone. It only exchanges HCO₃⁻ in the presence of Cl⁻. We did try to measure HCO₃⁻ current of WT pendrin in electrophysiology assays, but we could not make sure whether the current is caused by HCO₃⁻. The results were unreliable.

In the pendrin-HCO₃⁻ structure, the HCO₃⁻ interacts with the backbone nitrogen of S408 and L407, but not the side chains, so point mutation or point deletion may not be suitable. In addition, S408 and L407 missense variants and frameshift variants have been reported to be pathogenetic (PMID: 31599023, ClinVar ID: 285262, PMID: 22384008), indicating the importance of S408 and L407 in pendrin function.

8. Pendrin was purified in 100 mM NaCl buffer first and then 300 mM HCO₃⁻ is added before protein concentration for cryo-EM samples. Cl⁻ in the protein could be replaced by HCO₃⁻. The binding affinity between pendrin to Cl⁻ and HCO₃⁻ need to be measured in the condition of cryo-EM sample buffer.

Thanks for this suggestion. For *in vivo* experiment, we referred to the article of the mouse SLC26A9 structures (PMID: 31339488) for the anion conductivity sequence assay. However, as discussed in major question 5, we couldn't measure the Cl⁻ transport of pendrin properly. Neither could we measure that of HCO₃⁻ as discussed in major question 7. For *in vitro* experiment, the expression level and extracting rate of pendrin is not high, so we have not found an appropriate method to measure the affinity of pendrin.

9. The transport path of substrates in pendrin should be analyzed by HOLE.

Thanks for this suggestion. We supplied the analysis by HOLE. Since pendrin functions in the manner of alternative accessing, we analyzed the inward-open TMD and outward-open TMD, respectively. The figure below showed the cavities from the anion-binding pocket to the intracellular or extracellular space (Response Fig. 5). We added this new figure in Supplementary Fig. 4.

Response Fig. 5. Supplementary Fig. 4a The cavities from the anion-binding pocket to the intracellular or extracellular space (red cavity indicates the pathway diameter smaller than 3 Å, and green cavity indicates the pathway diameter larger than 3 Å).

10. Is there any direct interaction between Cα1b and transmembrane domain? How is its rotation related to the TMD conformation change?

Among all conformations we determined, only in inward-inward conformation, the helix Cα1b is close enough to have direct interaction with TMD (N579 of helix Cα1b could interact with D376 of TM8 in core region) (Response Fig. 6. a-c). In the asymmetric structures, the helix Cα1b of the outward-open protomer (blue colored in Response Fig. 6. e-f) rotates about 6° away and the core region moves far away up to the extracellular space. Either could break up the direct interactions. Furthermore, the missing IVS region (aa 596-650) is located right after helix Cα1b. IVS is too flexible to be captured in the density map, but it might have direct interaction with TMD. We added the figure below in Supplementary Fig. 5.

Response Fig. 6. a-c, Supplementary Fig. 5a-c In inward-open state C2 structures, the helix Ca1b is the most close to the TMD (N579 of helix Ca1b could interact with D376 of TM8 in core region). **d-f,** Supplementary Fig. 5d-f In the asymmetric structures, the helix Ca1b of the outward-open protomer rotates about 6° away and the core of the outward-open protomer moves up to the extracellular space. Therefore, below the inward-open TMD, N579 gets a little away from D376 by Ca1b rotation. And below the outward-open TMD, interaction is broken by core's movement.

Minor points:

11. line 99, No robust data in Extended data fig 1 supports the statement “pendrin forms a dimer as other family member”. The only related data is g panel, but the size exclusion chromatography is not enough to suggest “Pendrin forms a dimer”.

We agree that size exclusion chromatography is not enough to suggest “Pendrin forms a dimer”. During cryo-EM data processing, we only observed dimer particles. Thus, we revised this sentence to “Cryo-EM data processing showed that pendrin forms a dimer as other family members”.

12. line 175, “... pendrin is exposed a mixture ...” should be “pendrin is exposed to a mixture ...”

Sorry for the grammar errors, we corrected this sentence in the revised manuscript.

13. line 237, “despite the completeness differences”. This sentence is confusing.

Sorry for this confusion, we rewrote this section and removed the sentence in the revised manuscript.

14. line 244, “The starting residues 515-545 of STAS domain form a long loop region, however, it is well resolved due to the interactions with Cβ3, Ca5 and NTD”. It is recommended to remove

the word “However” from this sentence.

Sorry for the grammar errors, we corrected this sentence in the revised manuscript.

15. line 280, “significantly contribute to the initiation of anion transport within STAS domain”. There is no robust support for anion transport initiating from STAS domain. Citation should add if any paper supports this opinion.

Indeed, there is no robust support. In revision, we deleted this statement and rewrote this paragraph with citations as following: “STAS domains of SLC26 transporters have been hypothesized to serve as protein-protein interaction modules. The intracellular domain of Receptor tyrosine kinase Ephrin type-A receptor 2 (EphA2) is co-precipitated with pendrin in immunoprecipitation, and triggers internalization with pendrin (PMID: 32165640). Besides, GTPase activator and scaffold protein IQGAP-1 binds the STAS domain of hPDS/SLC26A4 and enhances its anion exchange activity (PMID: 35601831).” “Therefore, anion recruiting and/or intracellular protein signaling might induce conformation change of STAS domains, which in turn regulates the anion transport/exchange activities.”.

16. Fig 1b is not cited in the paper. Besides, topology should contain the information of cytosolic and extracellular side. It would be perfect to add this in the panel. The mapped mutations should cite the original papers.

Thanks for pointing out this. We now added the cell membrane labels in Fig. 1b and cited it in the revised manuscript. We also cited the original papers of the mapped mutations in the manuscript.

17. Fig 4f, S552 and S666 should be labeled more clearly.

Sorry for this confusion, we labeled them more clearly in the revised manuscript.

Reviewer #2 (Remarks to the Author):

SLC26A4 encodes an electroneutral anion exchanger, pendrin, and a large number of genetic variants associated with syndromic (Pendred syndrome, PDS) or nonsyndromic (DFNB4) hearing loss have been identified. Structural information of the pendrin protein is expected to greatly facilitate efforts in defining the pathological roles of these SLC26A4 variants. This study by Liu et al. reports cryo-EM structures of homodimeric mouse pendrin obtained under various anion substrate conditions. Overall, these pendrin structures look very similar to those recently reported by others for SLC26A5 (prestin) and SLC26A9. However, this study is novel in that an outward-open state was captured for the first time for SLC26 family of proteins. It is exciting that this outward-open structure was found in an asymmetric pendrin homodimer (outward-open + inward-open), which would provide mechanistic insights as to how electroneutral 1:1 anion exchange is mediated by the pendrin protein. I am enthusiastic about the cryo-EM pendrin structures reported in this paper; however, the manuscript contains multiple issues, especially in their functional assays, that need to be addressed. Please see below for my specific comments.

We appreciate the reviewer's work for reviewing our manuscripts, especially the comments and questions. Accordingly, we have revised our manuscript thoroughly and performed more assay as requested. The detailed response is appended in the following.

Major points

(1) Figs. 2e, 4d, and Extended Data Fig. 1. The authors tried to assess the anion transport functions of pendrin and its mutants by whole-cell voltage clamp. What is the rationale for using such an electrophysiological method for measuring pendrin's transport activity? It is my understanding that pendrin exchanges bicarbonate or iodide with chloride in a 1:1 electroneutral manner. Chloride/chloride exchange is also possible, but none of these exchanges would result in net charge movement (electroneutral). Why were the BCECF- and YFP-based optical methods (Fig. 5) not used throughout the manuscript?

Thanks a lot for this question. It was reported that pendrin showed sodium-independent transport of Cl⁻ and I⁻ in heterologous cell expression systems (*Xenopus laevis* oocytes or insect Sf9 cells, PMID: 10192399). Another study reported pendrin transports Cl⁻ and the transport of Cl⁻ depends on the membrane potential (whole-cell voltage clamp on COS-7 cells were performed, PMID: 15155570.). Referring to these studies, we carried electrophysiology assay on pendrin to test the function of mutants.

Your suggestion reminded us to test whether the current we measured by whole-cell voltage clamp is a chloride current. We used a broad-spectrum chloride channel inhibitor NPPB to measure the current change. NPPB is known as a blocker of chloride transport in a variety of different cells. NPPB can inhibit the current of pendrin on cells. It was reported that 0.1 mM NPPB inhibited the pendrin-dependent iodide influx by 33% (PMID: 27161422) and reduced the pendrin-chloride uptake by 59.3±8.4% (PMID: 16791000). Unfortunately, the addition of inhibitors did not significantly suppress the current, indicating that the current we measured were mainly generated by other ions instead of chloride. Therefore, we removed the electrophysiological assay data from the manuscript, as you suggested. As an exchanger, the anion exchange does not result in net

charge movement.

Then as you suggested, we performed BCECF- and YFP-based exchange assays to test the function of the key residues (Y105, S408, P142 and N457). We added these new results in main Fig. 5 and Supplementary Fig. 9 and 10.

(2) Fluorometric iodide/chloride antiport assay (Fig. 5f). I guess that fluorescence data were collected at a higher temporal resolution and Fig. 5f shows a summary of corrected fluorescence data only for four time points. Please show examples of raw fluorescence traces as well. Please also mention how the fluorescence data were corrected and what the error bars indicate (SD or SE). In this assay, it is important to confirm steady basal fluorescence ($\Delta F/\Delta t \sim 0$) before solution exchange and to determine the slope of fluorescence change ($\Delta F\%/\Delta t$). Note that some pendrin constructs (WT, G209V, and G672E) show large reductions in basal fluorescence even before solution exchange, and that the direction of fluorescence change after solution exchange is opposite to what is expected – increase in fluorescence upon iodide perfusion for F209V, G672E; no fluorescence recovery after perfusing back to a chloride solution for G209V, G672E, and E303Q. I am not convinced that this assay was done properly. Also, statistics needs to be provided for any comparisons.

Thanks for this question. In fluorometric exchange assay, confocal microscope was used to take photographs at four time points, so we only had raw micrographs without raw fluorescence traces. Because the fluorescence would be quenched with continuous exposure by laser, we referred to some articles with YFP-based assays and choose these four time points (PMID: 16914891, PMID: 15155570): After titration in wild-type pendrin transfected cells, we selected 140 mM Cl^- and 25mM I^- , as these concentrations tune the fluorescence intensity within the appropriate detecting range of fluorescence microscopy. Before exchange assay, cells were washed with fresh medium and incubated 10 min to make basal fluorescence steady ($\Delta F/\Delta t \sim 0$). When the bath buffer was changed from fresh medium to Cl^- -containing buffer for 5 min, the Cl^- -transport-induced fluorescence change had been steady; then when the bath buffer was changed from Cl^- -containing buffer to I^- -containing buffer for 2 min, the Cl^-/I^- -exchange-induced fluorescence change had been steady; at last, when the bath buffer was changed from I^- -containing buffer to Cl^- -containing buffer for 5 min, the Cl^-/I^- -exchange-induced fluorescence change had been steady.

As for data processing, we selected cells with both mCherry and YFP signals. In ImageJ, the selected cells in a series of micrographs were circled by a 20-pixel diameter circle and measured the mean fluorescence intensity (the circle is a little smaller than the cell). We defined the fluorescence intensity in medium after 10 min as F_0 and drew the $(F - F_0)/F_0$ curves ($n=5$, curve data points indicate mean, error bars indicate SD). In the figure below, from 0 min to 5 min, wild-type pendrin transfected cells showed a concentration-difference-induced Cl^- influx (as YFP fluorescence intensity reduced). Then from 5min to 7min, wild-type pendrin transfected cells showed an exchange-induced I^- influx (as YFP fluorescence intensity reduced further). At last, from 7 min to 12 min, wild-type pendrin transfected cells showed an exchange-induced I^- efflux (as YFP fluorescence intensity recovered). Then fluorescence intensity of YFP-transfected cells (negative control) fluctuated punily by endogenous Cl^- channels.

Response Fig. 7. Supplementary Fig. 9b, Curves of YFP fluorescence change, error bars indicate SD.

With the same program and microscope settings, we tested all pendrin mutants. During the first 5 min, G209V and G672E showed Cl⁻ transport activity like WT, while E303Q lost Cl⁻ transport activity like negative control and F335L showed large reduction of Cl⁻ transport. After treated with I⁻, G209V and G672E showed some fluorescence recovery that suggest the function loss of I⁻ transport and Cl⁻/I⁻ exchange (Because I⁻ could not be transported inward and Cl⁻ was transported outward when the bath buffer is no longer Cl⁻-containing, the fluorescence would recover). As for E303Q and F335L, after treated with I⁻, both showed fluorescence reduction. It suggested that E303Q and F335L still had I⁻ transport activity. As result, G209V, G672E and E303Q showed disability of Cl⁻/I⁻ exchange by losing Cl⁻ or I⁻ transport capacity, and F335L showed reduction of Cl⁻/I⁻ exchange compared with WT. These results were in agreement with the summary in the pendrin pathogenic mutations review (PMID: 27771369.).

Enlargement of raw fluorescence micrographs and un-paired t-test results are shown in Supplementary Fig. 9.

(3) Fluorometric bicarbonate/chloride antiport assay (Fig. 5g). A dual-excitation ratiometric pH-sensitive fluorescence probe, BCECF, was used; however, the authors monitored the fluorescence excited only with single excitation wavelength (488 nm). Consequently, the intracellular pH could not be determined. This is problematic, because the response of BCECF to pH is not linear, and because it is likely that the initial intracellular pH before solution exchange differs among the pendrin constructs. Thus, the data cannot be trusted in the present form. If a shorter wavelength light source were not available or if switching of excitation wavelengths were cumbersome, I would suggest to use a dual-emission ratiometric pH-sensitive fluorescence probe (e.g., SNARF, PMID: 31599023). Please show examples of intracellular pH traces (before/after solution exchange) along with a summary of transport activities ([H⁺]/sec). Again, statistics needs to be provided.

Thanks for suggestion of the probe SNARF with dual-emission and a better way to present fluorescence response with in-situ pH associated. However, technically, transfection with pendrin will increase cell mortality and the reduce cell adhesion in 293T cell line we used for experiments. In addition, because we don't have the automatic robot for buffer perfusion, the adherent cells will fall off significantly after three or four times of buffer exchange by pipette. Moreover, we use confocal microscope to measure fluorescence intensity, so taking multiple photographs at the same

area will cause strong fluorescence quenching. Another regular method for fluorescence intensity measurement is the microplate reader, but we did not successfully culture cells in 12-well plates or 24-well plates due to its low adhesion ability. Altogether, it is almost impossible for us to do in-situ pH associated fluorescence response so far.

Alternatively, we re-processed the data in an improved way (Response Fig. 8a and b). We selected cells with the same initial fluorescence intensity (Response Fig. 8c, there is no statistical significance between mutants and WT), to ensure that they have similar initial pH, consequently the pH change between experimental groups is more comparable. Thus, the new results are more convincing. We included these figures in main Fig. 5b and Supplementary Fig. 10.

Response Fig. 8. a and b, Supplementary Fig. 10c Curves of BCECF fluorescence change, error bars indicate SD. **c**, At 0 min, un-paired t-test results versus the cells transfected with WT pendrin. Error bars indicate SD.

(4) The authors should take advantage of the large number of PDS/DFNB4-associated SLC26A4 variants identified in patients to date and previous functional studies. For example, the presence of multiple disease-associated variants at P142, S408, and N457 (i.e., P142R, P142L, P142S, S408Y, S408P, S408F, N457K, N457D, N457Y, and N457I) implies that these missense variants are likely pathogenic, and previous functional studies demonstrated that P142R, P142L, S408F, and N457K indeed impair the function of pendrin (PMIDs: 18310264; 26752218; 31599023). Now, the pendrin structure reported in this study accounts for the underlying pathological mechanisms for these variants. I do not think that it is crucial to collect additional functional data for P142A, S408A, and N457A (Figs. 2e, 4d, and Extended Data Fig. 1) to claim the functional importance of these residues. In my opinion, what is more important is to facilitate efforts in predicting the potential pathological roles of variants with uncertain significance (VUS). Currently, ~800 SLC26A4 missense variants have been identified, approximately half of which are VUS (<https://deafnessvariationdatabase.org>). I strongly encourage the authors to expand Supplementary Table 3 by providing structure-based pathogenicity predictions for many more SLC26A4 missense variants rather than mostly duplicating previous experimental efforts (note that almost all deafness-associated SLC26A4 variants examined in Figs. 4 and 5 have already been characterized in previous studies).

Thanks for this suggestion. We detailly analyzed the disease-associated variants from a view of structure. Based on the deafness variation database, we listed all 761 missense variants

corresponding to 655 residues in our pendrin model in Supplementary Data (an excel) and add more analysis to the section of “Mapping of clinical disease-associated mutations”. Mutations are distributed in almost every region (Response Fig. 9, CL6 only contain 3 residues, aa 474-476). Therefore, we divided them into four groups: along the anion pathway, in the TMD cleft (interface between the core and gate region), in the hydrophobic periphery of TMD and in the dimerization interface at the STAS domain. Most disease-associated variants disturb the folding and affect the stability by changing the sidechain charge or introducing steric hindrance. We hope this Supplementary Data table could provide more information about the relation between pathogenicity and protein structure. We added these new figures in main Fig. 5 and Supplementary Fig. 11.

Response Fig. 9. Fig. 5i Location summary of 761 disease-associated mutations in the resolved 655 amino acids. Disease-associated mutations are cited from Deafness variation database (accessed on the 20 Sep 2022). NTD_{di} indicates the residue in the dimerization interface of NTD, EL indicates extracellular loop, CL indicates cytosolic loop, STAS_{di} indicates the residue in the dimerization interface of STAS.

(5) Fig. 3. How is the inward-open state found in the inward/outward asymmetric pendrin homodimer compared to those found in the inward-inward symmetric homodimer? Likewise, how is the outward-open state found in the inward/outward structure compared to those found in the outward-outward homodimer? Please describe if there are any noteworthy differences at any regions of the pendrin protein (e.g., N-terminus, dimerization interface, STAS, locations of the bound chloride and bicarbonate, etc.), as such information may provide significant mechanistic insights as to how conformation changes of the two protomers are coordinated to achieve 1:1 electroneutral antiport.

Thanks for this question. Superimposition of all 8 inward-open protomers from the 5 symmetric inward-inward and 3 asymmetric inward-outward conformations showed RMSD less than 0.5 Å, indicating that all inward-open conformations are very similar (Response Fig. 10a). The outward-open protomers from 1 symmetric outward-outward and 3 asymmetric inward-outward conformations are also very similar (Response Fig. 10b). We added these comparison figures in Supplementary Fig. 7. The RMSDs between different structures are updated in Extended Data Table 1.

Response Fig. 10. a, Supplementary Fig. 7a Superimposing the inward-open protomers in 8 pendrin structures (pendrin-Cl, pendrin-HCO₃, pendrin pendrin-Cl/HCO₃ ii, pendrin-Cl/I ii, pendrin-HCO₃/I ii, pendrin-Cl/HCO₃ io chain A, pendrin-Cl/I io chain A, pendrin-HCO₃/I io chain A). RMSD is listed in Supplementary Table 1. **b,** Supplementary Fig. 7b Superimposing the outward-open protomers in 4 pendrin structures (pendrin-Cl/HCO₃ io chain B, pendrin-Cl/I io chain B, pendrin-HCO₃/I io chain B, pendrin-Cl/HCO₃ oo). RMSDs are listed in Supplementary Table 1.

(6) Fig. 3b. Is it possible that just one bicarbonate ion binds to the outward-open state but with two distinct modes (e.g., one tightly bound and the other loosely bound)? Please describe in more detail.

Thanks for this question. We measured the distance between the two densities in the outward-open cavity and found that these two individual densities in the outward-open cavity are about 4.2 Å from each other (Response Fig. 11), which is too close to allocate two anions at the same time. Therefore, our previous description was incorrect. Instead, these two densities are artifacts coming from cryo-EM data processing, which combines two distinct binding modes of anion in the outward-open state. We added this new figure in Supplementary Fig. 2.

Response Fig. 11. Supplementary Fig. 2b Two anion densities in the outward-open cavity in pendrin-Cl/HCO₃. The distance between two densities is labeled.

(7) Extend data Fig. 3. How are anions bound to these structures? How are the inward- and outward-open structures compared to those found in other homodimeric structures (chloride/chloride, bicarbonate/chloride)? Extended data Table 1 indicates that there are differences, but readers would like to know where in the structures the differences are found.

Besides pendrin-Cl and pendrin-HCO₃, we observed the anion density in the inward-open pocket in 4 structures: pendrin-Cl/HCO₃_{io}, pendrin-Cl/I_{ii}, pendrin-Cl/I_{io} and pendrin-HCO₃/I_{ii} (Response Fig. 12). The positions of 4 anion densities are overlapped with Cl⁻ density in pendrin-Cl (Response Fig. 12a), but not HCO₃⁻ density in pendrin-HCO₃ (Response Fig. 12b). Eventually, they follow Cl⁻ binding coordination.

Response Fig. 12. a and b, Fig. 3d and f Details of the anion-binding pocket of pendrin-Cl and pendrin-HCO₃. Densities representing Cl⁻ and HCO₃⁻ are shown in light grey mesh. **c-f,** Supplementary Fig. 3 c-f The anion densities we absorbed in the inward-open pocket of the rest structures are shown in light grey mesh. Pendrin-Cl/HCO₃_{io}, pendrin-Cl/I_{ii}, pendrin-Cl/I_{io} and pendrin-HCO₃/I_{ii}.

As for the Supplementary Table 1, when superimposing different structures, all inward-open protomers are very similar with the RMSDs less than 0.5 Å. Similarly, all outward-open protomers are very similar with the RMSDs less than 0.5 Å. The RMSDs corresponding to the comparisons between inward-open and outward-open protomers are between 4.3 Å and 4.6 Å, and the major differences exist in the core region, as shown in main Fig. 3.

(8) Throughout the manuscript, avoid using the terms “pathogenic mutation(s)”. Instead, use “disease-associated variant(s)” or “deafness-associated variant(s)”.

Thanks for this suggestion. We corrected these terms in the revised manuscript.

(9) The manuscript contains many typos and grammatical errors that need to be fixed.

Sorry for the typos. The authors went through the manuscript carefully to correct them.

Minor points

(1) Line 98. “no other oligomers was observed”. I find a large elution shoulder preceding the pendrin peak in the SEC profile shown in Extended data Fig. 1g and smear above the pendrin band in Extended data Fig. 1h. What are they?

Sorry for this confusion. Confirmed by SDS-PAGE and negative stain results, the large elution shoulder contains pendrin aggregations varying in size and shape. It should be caused by instability after being extracted from the cell membrane. The blurry band above the pendrin appeared occasionally. We use that previous SDS-PAGE gel just because the pendrin band is ideal and compact. In fact, in most batches of purifications, pendrin is clean presenting no blurry band (Response Fig. 13). We replaced the SDS-PAGE gel to a new one showing purer pendrin in the revised Supplementary Fig. 1.

Response Fig. 13. SDS-PAGE gel of the peak fraction in different batches of purification.

(2) Line 123. “view of g showing ~” should be “view of c showing ~”.

Sorry for this typo. We corrected the legend in the revised manuscript.

(3) Figs. 2e and 4d and Extended data Fig. 1. The current data should be corrected for the cell membrane capacitance, as larger cells tend to express larger amounts of pendrin. Also, it is problematic that different statistical conclusions are derived for Y105A, P142A and S408F in Fig. 2e vs. Extended data Fig. 1d. In any case, I don’t think that this electrophysiological assay is appropriate for measuring pendrin’s transport activity. These data should be removed from the manuscript (see above).

Thanks for this suggestion. As mentioned in major question 1, we removed the electrophysiological assay data from the manuscript.

(4) Lines 147 and 158. Avoid saying “supposed to”, as it sounds highly subjective.

Thanks for this suggestion. We changed Lines 147 to “~because P142A mutation would loss pi-pi stacking between F141 and P142.” in Line 692. And we deleted the sentence in Lines 158 in the revised manuscript.

(5) Line 148. “~ will result ~” should be changed to “likely cause”.

Thanks for this suggestion. We changed this sentence in the revised manuscript accordingly.

(6) Lines 158-160. “~. Coincidentally, the allelic residue of pendrin P142 is Alanine in prestin and SLC26A9”. However, nonmammalian prestin orthologs and SLC26A9 have chloride transport activities, which is inconsistent with the authors’ argument here.

Thanks for this question, and sorry for the misleading. We rewrote this sentence in the revised manuscript as: “We measured the exchange function of P142A. Notably, P142A remains Cl⁻/I⁻ exchange function, but loses Cl⁻/HCO₃⁻ exchange function (Fig. 5g, h). This result is consistent with the fact that SLC26A9 is permeable to Cl⁻ but not HCO₃⁻.”.

(7) Lines 236-240. “In addition, the density of relatively stable cholesterol can be seen in all conformations, despite the completeness differences. Since cholesterol is believed to influence the localization and diffusion of prestin in plasma membrane, this may be characteristic for the interactions between SLC26 family members and plasma membranes”. What does “all conformations” mean? What is “completeness differences”? These sentences do not make sense.

Sorry for this confusion. “All conformations” means that we observed these lipid densities in all maps, including the inward symmetric maps, outward symmetric maps and asymmetric maps. And “completeness differences” means that the lipid densities in asymmetric maps are not as intact as them in the pendrin-Cl map (Response Fig. 14).

However, as the first reviewer suggested, we could not provide evidence whether these densities are cholesterol. Thus, we removed the cholesterol molecule from the model and deleted the discussion about cholesterol.

Response Fig. 14. Lipid densities between two TMD in pendrin-Cl, pendrin-Cl/HCO₃⁻_{io} and pendrin-HCO₃⁻/I_{io}.

(8) Line 276. Please show the locations of Y556, F667, and G672 in Fig. 4 (or in a supplementary figure).

Thanks for this suggestion. We added a new Supplementary Fig. 5 to show the locations of Y556, F667, and G672.

(9) Lines 278-281. Is the “positively charged platform” conserved among the SLC26 family? Please mention.

Thanks for this question. We analyzed the electrostatic potential surface of pendrin, prestin and SLC26a9 protomers, all showing that helix $\alpha 1b$ forms positively charged platform (Response Fig. 15). We mentioned this in the revised manuscript and added this in the new Supplementary Fig. 6.

Response Fig. 15. a-c, Supplementary Fig. 6a-c Electrostatic potential surface of pendrin, prestin and SLC26a9 protomers.

(10) Lines 297-303. “~ the side chain of leucine may disrupt protein-lipid interactions, ~”. The amino acid equivalent to F335 in pendrin is L325 in prestin. A lipid is found in close vicinity of L325 in the prestin structure (PDB: 7LGU), opposing the authors’ speculation.

F335L was reported as disease-associated variant that caused reduction of exchange activities (PMID: 19204907). Therefore, we suspected that leucine would not maintain the interaction as phenylalanine. Thanks for pointing out that L325 in prestin did not disrupt protein-lipid interactions. Without more evidence, we revised our previous deduction to “change the surface shape” in Supplementary Data table.

(11) Lines 308-309. “~ side chain of valine would increase steric hindrance of core-gate interface and contribute the positive surface charge”. How could valine contribute to positive surface charge?

Initially, we mutated G209 to Val in COOT, calculated the electrostatic potential surface in Chimera and observed a color change around the residue 209, indicating more positive charge. Therefore, we brought this point out. We agree that theoretically, the mutation between nonpolar amino acids is not likely to influence on the surface charge. We revised our previous deduction to “Steric hindrance surround; Interfere local folding” in Supplementary Data table.

(12) Line 311. Please define “pre-binding site” or cite a reference.

Thanks for this suggestion. The “pre-binding site” is reported by the article of rat prestin STAS domain crystal structures bound several anions. We cited this reference and supplied the figures in main Fig.4d and Supplementary Fig. 5.

(13) Lines 312-313. “In Cl⁻/I⁻ exchange assays, G672E lost I⁻ transport capacity, but maintains Cl⁻ permeability (Fig. 5f, j)”. The quality of the transport activity data is too low to determine the

functional consequence of G672E. Also, chloride transport data are not provided for G672E.

Thanks for the question. Similar as the explanation in major question 2, in the first 5 minutes, G672E could transport Cl^- like WT into the cell causing fluorescence quenching with decreased fluorescence intensity. The intensity changing is reliable and significant enough to support our conclusion, when the detailed explanation to the major question 2 is considered.

(14) Lines 336-338. Intracellular retention of the pendrin protein can be seen even for wild-type as evident in Figs. 5h and 5m. Thus, it is ambiguous what the authors mean by “with a well-defined cellular localization”.

We agree with this point. Since the micrographs we took were not clear enough to define the pendrin’s cellular localization, we removed a series of speculations about cellular localization.

(15) Lines 345-346. F335L may not affect lipid binding. See above.

Thanks for this question. We revised our previous deduction to “change the surface shape” in Supplementary Data table.

(16) Lines 356-357. “For R185T at the core-gate interface and L236P at the protein-lipids interface, the charge change may lead to local misfolding”. How could L236P affect charge?

Thanks for this question. We calculated the electrostatic potential surface of L236 and P236 by Chimera, showing that P236 introduced surface charge change. We agree that theoretically, the mutation between nonpolar amino acids is not likely to influence on the charge. So we revised the perspective “Interfere local folding of the α -helix” in Supplementary Data table.

(17) “The major difference is that the STAS domain of SLC26A9 has a distinct angular offset from pendrin and prestin, while the latter two basically overlap with each other”. Please explain this difference graphically (in a supplementary figure).

Thanks for this suggestion. We supplied a new figure in Supplementary Fig. 7 and added an arrow to point out the angular offset of SLC26A9 (Response Fig. 16).

Response Fig. 16. Supplementary Fig. 7c Superimposing the inward-open protomers of pendrin-Cl, prestin (PDBID: 7S9C) and SLC26A9 (PDBID: 6RTC). RMSDs are listed in Supplementary Table 2.

(18) Line 57 in Extended Data Fig. 6. “UniProt”, not “UniPort”.

Sorry for this typo. We corrected this spelling in the revised manuscript.

Reviewers' Comments:

Reviewer #1:

Remarks to the Author:

The authors have addressed most of the questions. However, there are still two points to be addressed.

1. The inverted alternate-access exchange model proposed in this work is not well supported by current results. The change of cross-section of one protomer does not necessarily need to be compensated by conformational change of the other protomer. Molecular dynamics simulation is suggested. It would be much convincing if the MD results indicates that the inverted conformation is a energy favoring state in Pendrin transport cycle.
2. The fluorescence experiment results are more reasonable in the revised manuscript. Both Y105F and Y105A show reduced activity of I⁻ and HCO₃⁻, supporting Y105 is vital in transport. However, it is necessary to check the protein behavior of these mutants, especially for Y105A, using size exclusion chromatography. The low transport activity could be a result of hampered protein folding or assembly.

Reviewer #2:

Remarks to the Author:

My major concern over the electrophysiological functional assay have been fully addressed. Specifically, using a pendrin blocker, the authors confirmed that the anion transport function of pendrin cannot be assessed by measuring whole-cell currents and removed the whole-cell data from the manuscript. The issues with the fluorometric functional assays have not been completely resolved; however, the reductions of the transport activity claimed for Y105A (Figs. 5g and 5h) and P142A (Fig. 5h) seem to be tenable, at least qualitatively, given the large differences in the fluorescence responses between variants vs. wild type. Since the main conclusion of this mostly descriptive report does not rely on the quality of the functional data that were collected for only a few selected variants, I would not dwell on the flaws in the fluorometric assay any further. The newly added supplementary file (Excel sheet) that summarizes structure-based pathological predictions is extensive and invaluable. Overall, the manuscript has been significantly improved. I only have minor comments for this revised manuscript.

Specific points

- (1) Line 96. "pendrin forms a dimer as other family members 20-23". Butan et al., 2022 (PMID: 35022426) and Futamata et al., 2022 (PMID: 36266333) also report homodimeric structures for prestin (SLC26A5). These references should also be added.
- (2) Ref #26 is cited multiple times between lines 147 and 157 as a reference for 18 pendrin missense variants. I believe that the authors meant to cite the deafness variation database (DVD) with Ref #26; however, DVD is not mentioned until line 353, giving a wrong impression that all the 18 variants were identified and reported in Ref #26. DVD should be introduced before line 147 to make it clear that Ref #26 is cited for DVD.
- (3) Lines 147-148. "S408F mutation almost abrogated transport activity 27". Dror et al., 2010 (PMID: 20442411) should also be cited here.
- (4) Line 150. "R409H showed reduction of exchange and transport function 28". Functional studies that actually measured the transport activity of R409H should be cited. Those are: Gillam et al., 2005 (PMID: 16053392); Wasano et al., 2020 (PMID: 31599023); Zhang et al., 2022 (PMID: 34545167).
- (5) Line 154. "P140H was reported to cause loss of Cl⁻/I⁻ exchange activity 29". Ref #29 did not determine the transport activity of P140H. Pera et al., 2008 (PMID: 19017801) should be cited (with the same lead author and both published in 2008, but these are two different papers).
- (6) Lines 157-158. "N457K reduced both Cl⁻/I⁻ and Cl⁻/HCO₃⁻ exchange 28". Ref #28 predicted the pathological effect of N457K. The functional consequence of N457K was experimentally determined by Wasano et al., 2020 (PMID: 31599023).
- (7) Line 196. "reveal two instinct binding modes". Do you mean "distinct" binding modes?

- (8) Line 246. "interactions with C β 3, Ca5 and NTD (Fig. 4c)". Please indicate C β 3 and Ca5 in the figure.
- (9) Lines 250-252. Are S552-S666 and R24-D724 interactions conserved among the SLC26 family members? Please mention.
- (10) Fig. 5i and Supplementary Fig. 11a. It would be probably better to include the location info (e.g., NTD, TM1, etc.) in the supplementary excel sheet.
- (11) Lines 328-330. "Y105F retained its Cl⁻/I⁻ exchange ability, although the transport activity of Cl⁻ and I⁻ decreased. However, the Cl⁻/HCO₃⁻ exchange activity of Y105F significantly decreased (Fig. 5g, h)". These decrements need to be claimed statistically. ANOVA followed by a multiple comparison test such as Tukey's or Dunnett's should be performed. Please include the statistics info in Fig. 5.
- (12) Line 363. "these regions contribute enormously to pathogenicity (Supplementary Fig. 9)". I think that "Supplementary Fig. 11a" should be referred to.
- (13) Line 378. "eventually causes a high pathogenicity rate (Supplementary Fig. 9)". I think that "Supplementary Fig. 11a" should be referred to.
- (14) Line 430. uniprot, not uniport.
- (15) Line 448. What is "PEI"? Please define or provide the product info.
- (16) Line 455. "digitonin power". Powder?
- (17) Supplementary Figs. 7a and 7b. Please provide the color-coding info as in panels c and d.
- (18) Supplementary Fig. 9. Please remove the data for S408A, N457A, G209V, G672E, E303Q, and F335L from the figure, as those are no longer referred to in the revised main text.
- (19) Supplementary Fig. 10. Please remove the data for S408A, N457A, E303Q, and F335L from the figure, as those are no longer referred to in the revised main text.
- (20) The manuscript contains multiple typos and grammatical errors that need to be fixed.

REVIEWERS' COMMENTS

Reviewer #1 (Remarks to the Author):

The authors have addressed most of the questions. However, there are still two points to be addressed.

1. The inverted alternate-access exchange model proposed in this work is not well supported by current results. The change of cross-section of one protomer does not necessarily need to be compensated by conformational change of the other protomer. Molecular dynamics simulation is suggested. It would be much convincing if the MD results indicates that the inverted conformation is an energy favoring state in Pendrin transport cycle.

Thanks for this suggestion. We employed the tech company for molecular dynamics simulation. Symmetric and asymmetric-inverted conformations were separately inserted in the equilibrated lipid bilayer, and the production run process was tested in 100 ns and 150 ns. We compared the final state with the initial state of each conformation, however, there were nearly no differences in RMSD and RMSF. Since they were captured by Cryo-EM, the symmetric and asymmetric-inverted conformations represent local energy minimum or global energy minimum. Transition from one conformation to the other may need to overcome a high energy barrier. Unfortunately, our current MD trials were not successful to prove which one is the energy favoring state.

2. The fluorescence experiment results are more reasonable in the revised manuscript. Both Y105F and Y105A show reduced activity of I- and HCO₃⁻, supporting Y105 is vital in transport. However, it is necessary to check the protein behavior of these mutants, especially for Y105A, using size exclusion chromatography. The low transport activity could be a result of hampered protein folding or assembly.

Thanks for this suggestion. We expressed and purified both pendrin-Y105A and pendrin-Y105F, as shown by the size exclusion chromatography assay, they both had the same aggregation shoulder and the same peak location as the wild type. Moreover, in the negative stain images, the mutants' particles had the same morphology as the wild type. In summary, we believed that neither Y105A nor Y105F would significantly hamper the assembly or folding of pendrin.

Reviewer #2 (Remarks to the Author):

My major concern over the electrophysiological functional assay have been fully addressed. Specifically, using a pendrin blocker, the authors confirmed that the anion transport function of pendrin cannot be assessed by measuring whole-cell currents and removed the whole-cell data from the manuscript. The issues with the fluorometric functional assays have not been completely resolved; however, the reductions of the transport activity claimed for Y105A (Figs. 5g and 5h) and P142A (Fig. 5h) seem to be tenable, at least qualitatively, given the large differences in the fluorescence responses between variants vs. wild type. Since the main conclusion of this mostly descriptive report does not rely on the quality of the functional data that were collected for only a few selected variants, I would not dwell on the flaws in the fluorometric assay any further. The newly added supplementary file (Excel sheet) that summarizes structure-based pathological predictions is extensive and invaluable. Overall, the manuscript has been significantly improved. I only have minor comments for this revised manuscript.

Specific points

(1) Line 96. “pendrin forms a dimer as other family members 20-23”. Butan et al., 2022 (PMID: 35022426) and Futamata et al., 2022 (PMID: 36266333) also report homodimeric structures for prestin (SLC26A5). These references should also be added.

Thanks for this suggestion. We added these references in the revised manuscript.

(2) Ref #26 is cited multiple times between lines 147 and 157 as a reference for 18 pendrin missense variants. I believe that the authors meant to cite the deafness variation database (DVD) with Ref #26; however, DVD is not mentioned until line 353, giving a wrong impression that all the 18 variants were identified and reported in Ref #26. DVD should be introduced before line 147 to make it clear that Ref #26 is cited for DVD.

Thanks for this suggestion. We introduced DVD “Around the anion-binding site, several disease-associated missense variants have been identified in patients and recorded in Deafness Variation

Database (<https://deafnessvariationdatabase.org>; accessed on the 20 Sep 2022)²⁹.” in Line 146.

(3) Lines 147-148. “S408F mutation almost abrogated transport activity 27”. Dror et al., 2010 (PMID: 20442411) should also be cited here.

Thanks for this suggestion. We added this reference in the revised manuscript.

(4) Line 150. “R409H showed reduction of exchange and transport function 28”. Functional studies that actually measured the transport activity of R409H should be cited. Those are: Gillam et al., 2005 (PMID: 16053392); Wasano et al., 2020 (PMID: 31599023); Zhang et al., 2022 (PMID: 34545167).

Thanks for this suggestion. We changed the cited review to these three references in the revised manuscript.

(5) Line 154. “P140H was reported to cause loss of Cl⁻/I⁻ exchange activity 29”. Ref #29 did not determine the transport activity of P140H. Pera et al., 2008 (PMID: 19017801) should be cited (with the same lead author and both published in 2008, but these are two different papers).

Sorry for this mistake. We changed the reference in the revised manuscript.

(6) Lines 157-158. “N457K reduced both Cl⁻/I⁻ and Cl⁻/HCO₃⁻ exchange 28”. Ref #28 predicted the pathological effect of N457K. The functional consequence of N457K was experimentally determined by Wasano et al., 2020 (PMID: 31599023).

Thanks for this suggestion. We added this reference in the revised manuscript.

(7) Line 196. “reveal two instinct binding modes”. Do you mean “distinct” binding modes?

Thanks for this question. Since we proved these two densities are artifacts coming from cryo-EM data processing, which combines two positions of anion in the outward-open state. We speculate that they are representing two moments of the anion transport process. We added this discussion in the revised manuscript.

(8) Line 246. “interactions with Cβ3, Cα5 and NTD (Fig. 4c)”. Please indicate Cβ3 and Cα5 in the figure.

Thanks for this suggestion. We corrected “Cβ3, Cα4 and NTD” and added labels in Fig.4 in the revised manuscript.

(9) Lines 250-252. Are S552-S666 and R24-D724 interactions conserved among the SLC26 family members? Please mention.

Thanks for this question. We checked the structures of prestin and SLC26A9 and the sequence alignment of all 10 SLC26 family members. As for S552-S666 interaction, it is at the main interface of two STAS domains. Although the residues are not conserved, high structural similarity allows SLC26 members to form interactions between two protomers around here. As for R24-D724 interaction, these two residues are conserved in most SLC26 members. We added this mention in the revised manuscript.

(10) Fig. 5i and Supplementary Fig. 11a. It would be probably better to include the location info

(e.g., NTD, TM1, etc.) in the supplementary excel sheet.

Thanks for this suggestion. We included the location information in the supplementary data excel.

(11) Lines 328-330. “Y105F retained its Cl-/I- exchange ability, although the transport activity of Cl- and I- decreased. However, the Cl-/HCO₃- exchange activity of Y105F significantly decreased (Fig. 5g, h)”. These decrements need to be claimed statistically. ANOVA followed by a multiple comparison test such as Tukey’s or Dunnett’s should be performed. Please include the statistics info in Fig. 5.

Thanks for this suggestion. We included the statistics information in the supplementary Fig. 9 and 10. In summary, One-way ANOVA with Dunnett’s multiple comparison test was performed for comparison between multiple groups. P value summary of multiple comparison results was labeled in histograms.

(12) Line 363. “these regions contribute enormously to pathogenicity (Supplementary Fig. 9)”. I think that “Supplementary Fig. 11a” should be referred to.

Thanks for this suggestion. We added this reference in the revised manuscript.

(13) Line 378. “eventually causes a high pathogenicity rate (Supplementary Fig. 9)”. I think that “Supplementary Fig. 11a” should be referred to.

Thanks for this suggestion. We added this reference in the revised manuscript.

(14) Line 430. uniprot, not uniport.

Sorry for this typo. We corrected this spelling in the revised manuscript.

(15) Line 448. What is “PEI”? Please define or provide the product info.

Thanks for this suggestion. We added this product information in the revised manuscript.

(16) Line 455. “digitonin power”. Powder?

Sorry for this typo. We corrected this spelling in the revised manuscript.

(17) Supplementary Figs. 7a and 7b. Please provide the color-coding info as in panels c and d.

Thanks for this suggestion. We added the color-coding information in the revised manuscript.

(18) Supplementary Fig. 9. Please remove the data for S408A, N457A, G209V, G672E, E303Q, and F335L from the figure, as those are no longer referred to in the revised main text.

Thanks for this suggestion. We removed this data in the revised manuscript.

(19) Supplementary Fig. 10. Please remove the data for S408A, N457A, E303Q, and F335L from the figure, as those are no longer referred to in the revised main text.

Thanks for this suggestion. We removed this data in the revised manuscript.

(20) The manuscript contains multiple typos and grammatical errors that need to be fixed.

Sorry for the typos. We went through the manuscript many times to correct any potential ones.